# Real-time measurements of NMVOCs in the central IGB, Lucknow, India: Source characterization and their role in O₃ and SOA formation

Vaishali Jain[1], Nidhi Tripathi[2], Sachchida N. Tripathi[1,3] Mansi Gupta[2], Lokesh K. Sahu[2], Vishnu Murari[1], Sreenivas Gaddamidi[1], Ashutosh K. Shukla[1], Andre S.H. Prevot[4]

[1]Department of Civil Engineering, Indian Institute of Technology Kanpur, Kanpur, 208016, India
[2]Space and Atmospheric Sciences Division, Physical Research Laboratory, Ahmedabad, 380009, India
[3]Centre for Environmental Science and Engineering, Indian Institute of Technology Kanpur, Kanpur, 208016, India
[4]Laboratory of Atmospheric Chemistry, Paul Scherrer Institute, 5232, Switzerland

*Correspondence to: Dr Sachchida N. Tripathi (snt@iitk.ac.in), Dr. Lokesh K. Sahu (Lokesh@prl.res.in )*

## Abstract:

Lucknow is the capital of India's largest state, Uttar Pradesh, one of South Asia's most polluted urban cities. Tropospheric photochemistry relies on non-methane volatile organic compounds (NMVOCs), which are ozone and secondary organic aerosol precursors. Using the proton-transfer reaction time of flight mass spectrometer (PTR-ToF-MS) at an urban background site in Lucknow, the chemical characterisation of NMVOCs was performed in real-time from Dec-2020- May 2021. About ~173 NMVOCs from m/z 31.018 to 197.216 were measured during the study period, including aromatics, non-aromatics, oxygenates, and nitrogen-containing compounds. The campaign daily mean concentrations of the NMVOCs were 125.5 ±37.5 ppbv. The NMVOCs daily average concentrations were about ~30% high during winter months (December-February) than in summer (March-May). The oxygenated volatile organic compounds and aromatics were the dominant VOC families, accounting for ~57-80% of the total NMVOCs concentrations. Acetaldehyde, acetone and acetic acid were the major NMVOCs species, 5-15 times higher than other species. An advanced multi-linear engine (ME-2) model was used to perform the NMVOCs source apportionment using positive matrix factorisation (PMF). It resolves the five main sources contributing to these organic compounds in the atmosphere. They include traffic (23.5%), two solid fuel combustion factors: SFC 1 (28.1%) and SFC 2 (13.2%), secondary volatile organic compounds (SVOC) (18.6%) and volatile chemical products (VCPs) (16.6%).  Aged and fresh emissions from Solid Fuel combustion (SFC 1 and 2) were the dominant contributors to total NMVOC, and compounds related to these factors had a high secondary organic aerosols (SOA) formation potential. Interestingly, the traffic factor was the second highest contributor to total NMVOCs and compounds related to this factor had high ozone formation potential. Significant differences in the composition of the two solid fuel combustion indicate the influence of local emissions and transport of regional pollution to the city. The high temperature during summer leads to more volatilisation of oxygenated VOCs, related to the VCPs factor.  The study is the first attempt to highlight the sources of NMVOCs and their contribution to secondary pollutants (SOA and O3) formations in Lucknow city during winter and summer. The insights from the study would help various stakeholders to manage primary and secondary pollutants within the city.

## 1. Introduction

Non-methane volatile organic compounds (NMVOCs) are carbon-containing gaseous compounds in the troposphere. NMVOCs can have significant effects (direct and indirect) on human health and the environment. These compounds have a half-life ranging from hours to months (Atkinson*, 2000). Exposure (Inhalation or direct contact) to high levels of NMVOCs can produce multiple chronic and acute health effects on humans, including nose, eyes, throat, and liver irritation. NMVOCs like benzene, acrolein, and aromatic amines are carcinogens subject to long-term exposure (WHO, 2021; Balakrishnan et al., 2015).The NMVOCs in the atmosphere act as precursors of ozone (O₃) and secondary organic aerosols (SOA) (Hallquist et al., 2009) They are oxidised by primary oxidant radicals such as hydroxyl radicals (OH), chlorine (Cl), and nitrate (NO3) in the presence of nitrogen oxides (NOx) and sunlight and can lead to the formation of ozone near the surface (Atkinson et al., 2004; Carter, 1994a) and also secondary oxygenated volatile organic compounds (OVOCs) (de Gouw et al., 2005). These OVOCs undergo further oxidation, gaining polar functional groups or oligomerise and becoming less volatile. When these compounds have sufficiently low vapour pressures, these products may condense to form a

particle-phase secondary organic aerosol mass (Hallquist et al., 2009; Heald et al., 2008; Monks et al., 2009. The
chemical composition of the parent compound, NOx concentrations and relative concentrations of OH and
chloride radicals during the day and $NO_3$ during the night (Warneke et al., 2004) are factors that ultimately
determine the fate of the formation of these aerosol products (Jang et al., 2002 et al., 2002). At high NOx levels,
VOCs degrade to form carbonyls, hydroxy carbonyls, organic nitrates and peroxyacetyl nitrates (PAN). In
contrast, low NOx conditions tend to produce fewer volatile compounds and organic peroxides after reaction with
HO2 radicals and favour SOA production from OVOCs.(Kroll et al., 2006; Ng et al., 2007; Hallquist et al., 2009;
Xu et al., 2014).
The inherent complexity in the non-linear VOCs-$NO_X$-$O_3$ relationship and change in ozone levels as a
function of VOCs and NOx is understood by ozone isopleths. When VOCs are relatively high and NOx is
relatively low, ozone production is limited by NOx, which is considered a NOx-sensitive regime. Conversely,
when VOCs are relatively low and NOx is high, ozone production is determined by the concentration of VOCs
and considered as VOC- sensitive regime (also known as a NOx-saturated regime) (Chameides et al., 1992). It is
observed that the urban area of Delhi was frequently associated with VOC-sensitive chemical regimes (Sharma
and Khare, 2017). The reduction of VOCs from anthropogenic emissions would reduce ozone levels more instead
of reducing NOx levels. (Sharma and Khare, 2017) also simulated that reducing NOx by 50% in Delhi would
increase ground-level ozone production by about 10-50%. In contrast, it is recommended that strategies control
abatement measures for NMVOCs, which would effectively reduce tropospheric ozone production by 60% more
than abatement of ozone or particulate matter ($PM_{2.5}$) alone. The buildup of Surface ozone and SOA
synergistically deteriorates the air quality and escalates harmful effects on humans and flora-fauna (Annenberg et
al., 2018; Burnett et al., 2014; Pye et al., 2021). The increased $PM_{2.5}$ concentrations and other pollutants lead to
economic and recreational loss, deterioration in the health of citizens, an increase in morbidity and premature
mortality risks, and biodiversity loss. Extreme haze events are one of the major challenges for Indian cities, being
among the most air-polluted cities in the world. Despite their importance, the spatial and temporal variability of
the concentrations of NMVOCs, which are precursors to secondary organic aerosols and ozone, remain unknown
in most Indian cities.
Only a few studies have observed and reported the ambient NMVOCs levels in Indian cities. These
studies are mainly conducted in large Indian cities such as Delhi (Garg et al., 2019; Hoque et al., 2008; Srivastava
et al., 2005; Tripathi et al., 2022), Mumbai (Srivastava et al., 2006), Kolkata (Majumdar et al., 2011;
Chattopadhyay et al., 1997, Sahu et al., 2016; Tripathi and Sahu, 2020; Sahu et al., 2017), Udaipur (Tripathi et
al., 2021; Yadav et al., 2019), and Mohali (Sinha et al., 2014). A previous study has presented the health risk
assessments for ambient VOCs levels in Kolkata (Chauhan et al., 2014; Majumdar (neé Som) et al., 2008). Most
of these studies have examined only a few NMVOCs, mainly (BTEX), with less or no information related to their
sources. Real-time characterization and source apportionment studies for NMVOCs in India are limited to the
national capital city of Delhi (Wang et al., 2020a; Jain et al., 2022; Stewart et al., 2021c), and Mohali (Pallavi et
al., 2019) across different seasons and sites. Traffic emissions and solid fuel combustion are observed to be major
contributors in both cities. Significant contributions from secondary VOCs are found in Delhi, while solvent-
based industries contributed to NMVOCs in Mohali. It is necessary to understand the different source profiles and
source contributions to ambient NMVOCs in different cities. The atmospheric interactions with radicals and
meteorology highly influence the concentrations of NMVOCs in the region. Recent source apportionment studies
based on real-time measurements of non-refractory fine particulate matter using HR-ToF-AMS identified various
sources present at different sites in Delhi (Lalchandani et al., 2021; Shukla et al., 2021; Tobler et al., 2020). These
studies emphasized that it is essential to understand the variance of sources between day-to-night and different
seasons. The significant contributors to fine suspended particulate matters in the National Capital Region are the
burning of crop residues in neighboring states and open burning of waste, as well as the increased construction
activities, industrial expansion, thermal power plants, number of vehicles (two-wheelers and cars), and residential
fuel use that result from an ever-increasing population. In addition, recent studies based on real-time
measurements of NMVOCs using PTR-ToF-MS in Delhi (Wang et al., 2020a; Jain et al., 2022) and Mohali
(Pallavi et al., 2019) emphasized the importance of source characterization of NMVOCs simultaneously. Very
few source apportionment studies highlighted the sources of NMVOCs present in other Asian cities (Wang et al.,
2021a; Tan et al., 2021; Fukusaki et al., 2021a; Sarkar et al., 2017; Hui et al., 2018). These studies highlighted
that NMVOCs sources have substantial value in checking the secondary aerosols formation and air quality.
The lack of identification of the sources and relative contribution of NMVOCs remains a challenging task for
policy-driven measures. The development and evolution of strategies need an understanding of the seasonal and

temporal variations and sources of NMVOCs. The reaction pathway is different for different NMVOCs and depends on the reaction rates of the species. Therefore, the ozone formation potential (OFP) of all NMVOCs is not the same. NMVOCs are categorised into distinct families based on their chemical structure and mass/charge (m/z) ratios. Some of these NMVOCs have lower OFPs than others and tend to form less ozone in the atmosphere. Understanding these OFPs in different chemical regimes would help identify families or species of NMVOCs of greater concern for surface ozone production control.

Here, in this study, a real-time instrument, PTR-TOF-MS (Proton Transfer Reaction Time of Flight Mass Spectrometer), is deployed for a period of 112 days (Dec-May) in Lucknow city to understand the contribution of long-range transport and local VOC emissions. Lucknow, also known as the 'City of Nawabs', is an urban city situated in the centre of the Indo-Gangetic Basin region on the banks of the Gomati River. It is one of the fastest-growing cities and is now known for its manufacturing, commercial and retail hub. The exploding population due to increased migration from nearby towns and villages have widened the city boundaries. Currently, the city has two major Indian National Highways (NH-24 and NH-30) interjecting. The city has 125 petrol/diesel filling stations and seven designated industrial areas (Anon, 2018). The number of registered personal motor vehicles in the city as of 2017 is about ~2 million (Government of India, 2019), that had been increasing at an average rate of 9% every year since 2007. Besides this, 255 brick kilns operate within and around Lucknow city. Only ~4.7% of the area of the district is covered by forest area with ~ 2.8 million population (Census of India, 2011). The city has eight large-scale, public-sector undertakings, eleven medium-scale industries, and hundreds of micro, small and medium enterprises (MSMEs) (Anon, 2018). Increased industrial and construction activities, unregulated energy and fuel consumption, unchecked vehicular pollution and unsustainable urbanisation are major driving forces for poor air quality in Lucknow (Uttar Pradesh Pollution Control Board, 2019). The aerosol loadings in the city have been unprecedentedly high for the last two decades (Sharma et al., 2006; Markandeya et al., 2021; Lawrence and Fatima, 2014). The $PM_{2.5}$ concentrations were found to be highest in the industrial area during winter compared to residential and commercial spaces (Pandey et al., 2012, 2013). Nevertheless, minimal work has been conducted to investigate air pollution and its health impacts in the city, most of which are focused on particulate pollution. To our knowledge, there are no reported measurements of NMVOCs over the city.

This study discusses the first-time-ever measurements of NMVOCs using PTR-ToF-MS over the crucial site in the middle of the Indo-Gangetic basin (IGB). This study focuses on the relative contribution of the different sources of NMVOCs using positive matrix factorization (PMF) and their associations with organic aerosols. Recently developed and extensively used receptor Model, PMF for source apportionment studies can identify physically relevant environment factors more robustly than other models(Paatero and Tapper, 1993, 1994). The present study also studied the influence of meteorological parameters such as temperature, relative humidity, and solar radiation on the diurnal and seasonal variation of NMVOCs. A specific goal of this study is to distinguish between primary emissions and secondary formations of NMVOCs. Moreover, the contribution of different sources of NMVOCs towards ozone and secondary organic aerosol formation is also estimated. The key highlight of the study is its comprehensive coverage of about 173 species of NMVOCs in Lucknow city for two seasons. From our knowledge, 173 different species of NMVOCs have not been reported elsewhere in India. The insights from the results of the study would help the authorities channel the strategies for controlling NMVOCs and forming secondary pollutants (Ozone and SOA).

## 2. Methodology

### 2.1. Sampling site description

The sampling site (26° 51' 55.4" N, 81° 0' 17.5" E) is marked as a red triangle in Figure 1. It is located in Lucknow city at a height ~12 m above the ground of the Uttar Pradesh Pollution Control Board (UPPCB) office building in Gomti Nagar. Residential buildings, office complexes, schools, big parks and commercial spaces surround the sampling site. The industrial and manufacturing plants within and around the city are related to steel metal components and fabrication, automobile parts, chemical industries, and food and agro-based and handicraft sectors (chikankari, zardozi, bone craft). The Industrial map of Lucknow and nearby districts with major/mini-industrial areas, large/medium scale industries, sewage treatment plants, solvent-based industries, sugar mills, pharmaceutical industries, and power plants are also shown in Figure 1. The measurements of NMVOCs were conducted using a proton-transfer-reaction time-of-flight mass-spectrometer (PTR-TOF-MS, Ionicon Analytik GmbH) from 18th December 2020 to 5th May 2021, covering winter and summer seasons. The study period is divided into two seasons according to the classification by IMD (Indian Meteorological Department) as winter (Dec-Feb) and summer (March-May). The gaps in the sampling period from 3-8th January and 21st March - 9th

April were due to maintenance and calibration of the instrument. The average daily temperature was ~28 ºC over
the whole study period in the city. The mean daily temperature during winters (Dec-Feb) was around ~25±2.5 ºC
and during summers (March-May) around 32±3 ºC. The relative humidity ranged from 64±14% during winters
and 42±11% during summers. The comparison of temperature and relative humidity changes during both seasons
are shown as box plots in supplementary Figure S1. These values are based on the days when NMVOCs
measurements exist. The pre-dominant wind direction was South-Southeast during colder and Southwest during
warmer periods, as shown in supplementary Figure S1. The wind speed is relatively calm during winters than in
summers. All the instruments were placed inside a temperature-controlled laboratory during the campaign.
Detailed descriptions of the instruments can be found in subsequent sections.

## 2.2. PTR-ToF-MS measurements of NMVOCs

The PTR-TOF-MS is widely used for measuring NMVOCs with high mass resolution and sensitivity. A detailed
description of the instrument can be found in other studies (Jordan et al., 2009; Graus et al., 2010; Tripathi
and Sahu, 2020; Tripathi et al., 2022), while a brief description is given here. The PTR-TOF-MS is based on
the chemical ionisation method, facilitated by proton-transfer reactions with hydronium ($H_3O^+$) ions as the primary
reactant ion, which causes much less fragmentation of organic molecules in the sampled air. The natural
components of air (nitrogen, oxygen, hydrogen, carbon dioxide, Argon) have a lower proton affinity than water
molecules. They thus do not react with $H_3O^+$, while most VOCs have higher proton reactivity than water,
facilitating non-dissociative proton transfer. These $H_3O^+$ ions are generated with high efficiency (~99.5%) through
a hollow cathode discharge source, and then these reactant ions enter the adjacent drift tube section. The sampled
air is also injected into the drift tube section, where proton transfer reactions between hydronium ions [$H_3O^+$] and
neutral VOCs [$R_iH_j$] occur to form protonated product VOC ions [$R_iH_{j+1}$], and water molecules [$H_2O$] as shown
in equation 1. These [$R_iH_{j+1}$] then enter the orthogonal acceleration reflectron time-of-flight mass spectrometer
via a specially designed transfer lens system (Jordan et al., 2009).

$$H_3O^+ + R_iH_j \rightarrow R_iH_{j+1} + H_2O \tag{1}$$

The parameters of the drift tube of the instrument were maintained at 2.2-2.4 mb, 60º C, 600 V, 130 Td
for pressure, temperature, voltage and electric field (E/N; where E = electric field strength and N= gas number
density) respectively and operated with a time resolution of 30 seconds. Typically, these or similar values have
been observed as most suitable for ambient air measurements of NMVOCs (Blake et al., 2004, 2009). During the
study, PTR-ToF-MS instrument's inlet was connected to a Teflon PFA (perfluoroalkoxy) tube (1.5m in length)
for drawing air samples at the flow rate of 60mL/min. The inner diameter of the tube was 0.75mm, and the
residence time of the air in the inlet was less than 1 sec. The PTR-ToF-MS can identify hydrocarbons (HC) and
oxygenated VOCs at sub-ppb levels within a second (Graus et al., 2010; Müller et al., 2012). In this study, the
PTR-ToF-MS measured 173 NMVOCs (m/z 31.018 to 197.216) at the sampling site. The reaction rates (k) of the
ions were applied from the literature (Cappellin et al., 2012). "A rate constant of $2 \times 10^{-9}$ cm$^3$ s$^{-1}$ was assumed
for all ions whose reaction rates (k) were not available in the literature (Smith and Spanel, 2005). The overall
uncertainties were in the range of 8%−13% in the calculations of the mixing ratios of VOCs present in the standard
mixture. The cause of uncertainties in calculating VOC mixing ratios includes the uncertainties in the mass flow
controllers (MFCs) of GCU and standard mixture (±5%−6%). The reaction rates (k) of the ion were applied from
the literature (Cappellin et al., 2012). A rate constant of $2 \times 10^{-9}$ cm3 s−1 was assumed for all ions for which
reaction rates (k) were not available in the literature. (Hansel et al., 1999; Steinbacher et al., 2004) have reported
up to 30% of uncertainty in the calculations of the mixing ratios of VOCs due to the k reaction rate. The calibration
of the instrument was performed at the starting, middle and end of the campaign using a certified standard gas
mixture (L5388, Ionicon Analytik GmbH Innsbruck, with a stated accuracy better than 8%) containing ~1.0 ppmv
of VOCs. A detailed description of the calibration setup and details, including zero measurements, is given in the
previous studies (Jain et al., 2022; Tripathi et al., 2022), and in the supplementary Figure S2. For method detection
limits (MDL), we have calculated the MDL using 3σ (standard deviation) of the zero air of 20 min time duration
data points. The exact mass-identified chemical formula and family of species of the observed NMVOCs (173 in
number) is given in supplementary Table S1. The mixing ratios of these measured 173 NMVOCs are averaged
over the study period and compared in the box plots as shown in Supplementary Figure S3. The three most
abundant NMVOC species are observed as acetaldehyde, acetone, and acetic acid.

## 2.3. HR-ToF-AMS measurements of NR- PM$_{2.5}$

A high-resolution time-of-flight aerosol mass spectrometer (HR-ToF-AMS, Aerodyne Research Inc., USA) was also deployed for campaign measurements. HR-ToF-MS (Decarlo et al., 2006) measures size-resolved mass spectra of non-refractory PM$_{2.5}$ (NR- PM$_{2.5}$) with high time resolution (2 mins). A detailed description of the instrument can be found in other studies (Lalchandani et al., 2021; Shukla et al., 2021) and is explained briefly here. The ambient aerosol particles were sampled through the PM2.5 cyclone (BGI, Mesa Labs, Inc.), which gets transmitted through stainless steel tubing (~ 8 mm inner diameter and ~10 mm outer diameter) with a maintained flow (0.08 lpm). This setup is further connected to a Nafion dryer (MD-110-144P-4: Perma Pure, Halma, UK) to reduce moisture content and then connect to the instrument's sampling inlet. The ambient aerosols enter the aerodynamic lens through a sampling inlet (100 µm diameter critical orifice) and focus on a narrow beam. This particle beam then enters a sizing chamber, where it can be sorted based on its size. This size-resolved beam enters the vaporisation chamber, and the non-refractory part of the particles (Nr-PM$_{2.5}$) vaporises at 600º C and ~$10^{-7}$ Torr. These gaseous molecules are then ionised and detected by a ToF-MS, depending on their m/z ratio. HR-ToF-AMS was operated in the high sensitivity V-mode for two cycles of the 60s (total 2 minutes), regularly switching between MS and PToF mode for 30 seconds each. During the study period, the particles-free air was provided for 1-2 hours every week to check and correct the fragmentation table at m/z's 12, 16, 18, 29, 33, 40, 44. The IE (Ionization efficiency) calibrations were performed at the beginning, middle and end of the campaign study following the mass-based method (Jayne et al., 1998, 2000) with an SMPS (scanning mobility particles sizer) unit (TSI Inc.). The raw data from HR-ToF-AMS was analysed for unit mass resolution (UMR) and high resolution (HR) using SQUIRREL (version 1.59) and PIKA (version 1.19) toolkit in Igor Pro software (version 6.37). The NR-PM$_{2.5}$ is chemically characterized into organics (Org), nitrates (NO$_3$), sulphates (SO$_4$), and chlorides (Cl). The organic aerosols mass spectra obtained from HR analysis and UMR analysis were combined from m/z 12 to 300 (~422 ions) to make the input matrix for PMF (positive matrix factorization) analysis. The PMF analysis was performed using the ME-2 engine implemented in SoFi Pro (Source Finder, Datalystica Ltd., Switzerland) (Canonaco et al., 2013, 2021) in a graphical interface software Igor Pro version 6.37 (Wavemetrics, Inc., Portland). The detailed analysis and results of PMF of NR- PM$_{2.5}$ are given in other studies (Lalchandani et al., 2021; Tobler et al., 2020; Talukdar et al., 2021), beyond the scope of this paper. In brief, the organic aerosols (OA) mass spectra from HR-ToF-AMS were explained by 5-factors consisting of one hydrocarbon-like organic aerosols factor (AMS_HOA), two solid fuel combustion factors (AMS_SFC/BB& AMS_SFC/OA), one more-oxidised oxygenated OA (AMS_MO-OOA) and one low-oxidised oxygenated OA (AMS_LO-OOA).

## 2.4. Supporting measurements

An Aethalometer (Magee Scientific, model AE-33) was also deployed at the campaign site to measure the real-time black carbon (BC) mass concentrations. It collects the aerosol particle samples on the quartz filter tape and quantifies the optical attenuation at seven different wavelengths (370, 470, 520, 590, 660, 880 and 950 mm) with high temporal resolution (1 min). It is based on a dual-spot technique for loading corrections (Drinovec et al., 2015). The change in optical attenuation measurements in the selected time interval at 880nm is converted to equivalent BC measurements (eBC) using the mass absorption cross section (MAC) of 7.77m$^2$ g$^{-1}$ (Drinovec et al., 2017, 2015). Using the enhanced absorption of biomass-burning aerosols in the near ultra-violet and blue wavelength range, the Aethalometer's multi-wavelength BC data may be apportioned into biomass burning and traffic combustion sources(Sandradewi et al., 2008; Zotter et al., 2017). The model employs an absorption ngströmexponent (AAE) value that corresponds to both vehicular and biomass combustion as the primary source of light-absorbing particles. In this study, the AAE value of 0.9 for traffic and 1.5 for biomass burning emissions is based on previous studies (Tobler et al., 2020; Lalchandani et al., 2021). More details about the instrument can be found in the previous studies (Lalchandani et al., 2021; Shukla et al., 2021). The sampling site (building) is a part of the national central ambient air quality monitoring stations (CAAQMS). The meteorological parameters (temperature, relative humidity, wind parameters) and concentrations of trace gases (NO$_2$, SO$_2$, Ozone) are downloaded from the CAAQMS dashboard (https://app.cpcbccr.com/ccr/#/caaqm-dashboard-all/caaqm-landing) managed by the central pollution control board (CPCB), the government of India for Gomti Nagar station, Lucknow.

## 2.5. Source apportionment

Numerous receptor models have been used to analyse the dynamic behaviour of ambient aerosol measurements and relate it to physical sources. One of the recently developed algorithms, positive matrix factorisation (PMF) (Paatero and Tapper, 1994), has been explored by numerous studies to apportion the measured bulk composition and temporal variation of aerosols (Zhang et al., 2011; Talukdar et al., 2021). The PMF algorithm is a non-

negative, symmetrical factor analytic technique that produces unique factorisation by iterative reweighting of individual data values and unique solutions. It solves the common bilinear equation, given as (Eq. 2):

$$X\ (mxn) = G\ (mxp)\ F(pxn) + E \tag{2}$$

where X represents the measured matrix, G and F are unknown matrices, and E is the error/ residual matrix. The m and n represent the time series and individual mass dimensions, and p is the number of factors. The calculated quantities G and F represent timeseries and profiles of the specific factor of the model solution, respectively. The ME-2 solver decreases the rotational ambiguity and fits the G and F entries to minimise the uncertainty in quantity 'Q'. This 'Q' is the sum of the squared residuals weighted by their respective uncertainties, as given in equation 3. From the equation, it can be inferred as a normalised chi-square metric, where $e_{ij}$ represents the residual matrix of E and $\sigma_{ij}$ represents measured data uncertainties.

$$Q = \sum_{i=1}^{m} \sum_{j=1}^{n} \left(\frac{eij}{\sigma ij}\right)^2$$

$$Q_{exp} = n.\,m - p.\,(m + n) \tag{3}$$

Another quantity, $Q_{exp}$, degree of freedom, depends on the dimensions of the matrix and the number of factors. In an ideal case, the ratio of $Q/Q_{exp}$ is expected to be 1, with all the elements of the measured matrix and uncertainties well-defined. However, it has been noticed in earlier studies that the absolute value of the ratio, $Q/Q_{exp}$, is not always equal to 1 due to errors in measured data uncertainties, transient sources, and unknown model residuals. It is recommended to use relative change in this ratio and characteristics of the physical source while choosing the optimum factor solution (Paatero and Tapper, 1993, 1994). In this study, this algorithm is applied over the measured NMVOCs mass spectra using ME-2 (multi-linear engine) (Paatero, 1999) over SoFi Pro (Source Finder, Datalystica Ltd., Switzerland) (Canonaco et al., 2013) in a graphical interface software Igor Pro version 6.37 (Wavemetrics, Inc., Portland). Earlier studies have applied a similar PMF algorithm over mass spectra of 90 NMVOCs in Delhi (Jain et al., 2022; Wang et al., 2020a) and 101 NMVOCs in Beijing (Wang et al., 2021a). In this study, for the first time, we have included 170 NMVOCs measured by PTR-ToF-MS from m/z 42.034 to m/z 197.216. The input and residual error matrix for the PMF analysis were prepared using timeseries of mass spectra and calculated individual errors for each data point, as explained in the previous study (Jain et al., 2022). After incorporating the calibration factors, the uncertainties or residual error matrix are estimated by multiplying the peak area with the correction matrix. The total uncertainties vary in the range of 8-12% during calculations of the mixing ratio of NMVOCs. The three most abundant NNMVOCs are not included in the PMF analysis due to their high signal-to-noise ratios and relatively higher (about 5-15 times) concentrations than other NMVOCs, as shown in supplementary Figure S3. The pretreatment of the input matrix also includes applying a minimum error threshold. The weak variables, having a signal-to-noise ratio <2 and bad variables, having a signal-to-noise ratio <0.2, are down-weighted by 2 and 10, respectively (Paatero and Hopke, 2003; Ulbrich et al., 2009).

The PMF algorithm calculates factor profiles, unlike the chemical mass balance (CMB) receptor model. The most crucial decision for the interpretation of the findings of the PMF is selecting the optimum modelled number of factor solutions. This is achieved by applying several mathematical metrics, correlating with external measurements, and interpreting the physical sources. The ratio of $Q/Q_{exp}$ is first examined for every factor solution. The factor solution having an absolute value of $Q/Q_{exp}$ ratio near 1 indicates an accurate estimation of errors, and it should be selected but not observed for real observations. The $Q/Q_{exp} \gg 1$ and $\ll 1$ indicate under and overestimation of errors or variability in the factor solution, respectively. It is anticipated that Q will drop with each addition of the number of factors, as this introduces extra degrees of freedom to improve the fit of the data. Another important metric is the evaluation of scaled residuals in the timeseries and mass spectra. The scaled residuals ±3 for each data point in the time series are considered evidence of a good solution (Paatero and Hopke, 2003; Canonaco et al., 2021). The supplementary Figure S4 shows the scaled residuals over the timeseries and diurnal cycle for the 3-10 factor solution. In the present study, the $Q/Q_{exp}$ does not lie near 1, but the high % change in $Q/Q_{exp}$ is observed while examining 3-5 factor solutions, as shown in Figure 2. The total scaled residual of all species is calculated and plotted for different factors in Figure 2. The changes in the residuals and the drops in Q/Qexp indicate that the 5-factor solution is an optimum solution. This solution is further analysed regarding their mass spectral features, time series and correlation with external tracers (Org, NO₃, SO₄, Cl from Nr-PM$_{2.5}$, organic resolved factors, gases (O₃, NO, NO₂, NOx, SO₂), temp, RH, WD, WS, and BC concentrations).

The optimum factor solution from the PMF analysis was further refined by self-constraining the
secondary volatile organic compounds (SVOC) factor with random values varying from 0.1 to 1 with delta a= 0.1.
Finally, a = 0.3 was chosen as the optimum solution after examining the temporal and diurnal variation of the
factor. More details about the constraining of the solution are explained in supplementary text ST1. Further, The
uncertainty of the selected solution is quantitatively addressed by bootstrap analysis (Davison and Hinkley, 1997;
Paatero et al., 2014), a module available in the SoFi Pro (Canonaco et al., 2021), as explained in supplementary
text ST2. Previous studies have also followed this methodology for uncertainty estimation of organic aerosols
source apportionment (SA) results (Lalchandani et al., 2021; Tobler et al., 2020; Shukla et al., 2021; Lalchandani
et al., 2022), elemental aerosols SA results (Shukla et al., 2021), and VOCs SA results (Wang et al., 2021a; Jain
et al., 2022; Stewart et al., 2021c). The uncertainty or $PMF_{error}$ is observed as 1% or less for all factors
Supplementary Figure S5. This infers that the 5-factor solution is a statistically robust solution with rather low
uncertainty.

**2.6. Ozone formation potential and SOA yield of NMVOCs**
Ozone formation potential (OFP) is a reactivity-based estimation technique to assess the sensitivity of the
VOCs for ozone formation (Carter, 2010, 1994b). Numerous VOCs are emitted into the atmosphere from various
sources, followed by distinct reaction pathways and have different OFPs. The calculated reactivities of VOCs
have been investigated in multiple modelling studies depending on the environmental conditions (Carter, 1994a).
This approach is based on calculating OFP using maximum incremental reactivity (MIR) values for individual
VOC species, reported and updated by (Carter 2010). MIR values are calculated as the change in the ozone formed
by adding a VOC to the base case in a scenario with adjusted NOx concentrations. OFP of individual VOCs is
estimated using Equation 4. Here, this equation is adopted (Carter, 2010, 1994b) and modified for this study to
calculate the ozone formation potential for each of the factors, resolved from PMF analysis as given below (Eq.
336 5),

$$OFP(j) = [VOC_j] \times C_j \times MIR_j \tag{4}$$

$$OFP(i) = \sum_{j=0}^{n}[VOC_j] \times C_j \times RC_{ji} \times MIR_j \tag{5}$$

Where, $OFP(j)$ and $OFP(i)$ represents the ozone formation potential for an individual VOC (j) and a factor
number (i) , respectively, expressed in $\mu g/m^3$. $[VOC_j]$ represents the mixing ratio (ppbv) of a given VOC ion (j),
$C_j$ is the number of carbon atoms present in each VOC ion (j), $RC_{ji}$ is the relative contribution of VOC ion (j) to
the factor (i). $MIR_j$ is the maximum incremental reactivity of a $VOC$ ion (j). The $MIR_j$ are adopted from the
(Carter, 2008, 2010, 1994a). The above equation is used to compute the OFP for each factor and determine which
source factor contributes the most to ozone generation, as explained later. The MIR values are available for a
limited number (40) of NMVOCs, given in supplementary Table S2. The NMVOCs without reported MIR values
are not considered for OFP estimation.
The chemical pathways and reaction products involved in SOA formation from NMVOCs are poorly
understood or even unknown. Estimation of SOA formation (SOA yield) has been largely constrained to indirect
methods due to the complexity of the chemical matrix of organic aerosols and the lack of direct chemical analysis
methods. Numerous studies have estimated SOA yield from different species involving computer modelling and
chamber experiments (Zhang et al., 2017). The smog-chamber studies help estimate the value for SOA yield is
more reliable as they mimic the actual scenarios. These parameters also helped in improving model
parameterization and SOA mitigation strategies. For the current study, the contribution of an individual NMVOCs
species to SOA is estimated by multiplying the SOA yield by the concentration of the NMVOC species in the
atmosphere (amount available for the reaction), as shown in Equation 6. The SOA yields $Y_{SOA\,(j)}$ reported by
Bruns et al 2016 were used for this analysis. The compounds for which SOA yield values are not available from
the literature directly, it is estimated that compounds having carbon atoms more than 6 (C >6) will have the same
SOA yield of 0.32 (Bruns et al., 2016). Based on their structure, the compounds (C >6) are considered to contribute
significantly to SOA (Bruns et al., 2016). The average value (0.32) of the published SOA yield of 18 compounds
(C>6) is used. In this study, the individual SOA yield values considered are given in the supplementary Table S3.
The contribution of the individual factor to SOA formation is also estimated using (Eq. 6 and 7) as given here.

$$C_{SOA}(j) = Y_{SOA(j)} \times VOC_j \qquad (6)$$

$$C_{SOA}(i) = \sum_{j=0}^{n} VOC_j \times RC_{ji} \times Y_{SOA(j)} \qquad (7)$$

Where, $C_{SOA}(j)$ and $C_{SOA}(i)$ represents the contribution to SOA formation for and individual VOC (j) and a factor number (i) respectively, expressed in $\mu g/m^3$. $VOC_j$ represents the concentration ($\mu g/m^3$) of a given VOC ion (j), $RC_{ji}$ is the relative contribution of VOC ion (j) to the factor (i). $Y_{SOA(j)}$ is the SOA yield of a $VOC$ ion (j). This analysis represents the estimated OFP and SOA formation potential of the air mass composition at the sampling site, not the actual OFP and SOA formation potential from various sources. This means that airmasses dominated by fresh emissions (e.g., traffic) will have a different OFP and SOA formation potentials than those in aged airmasses (e.g., long-range transport of BB plumes) or any other source.

### 2.7. Concentration weighted back trajectory analysis

Concentrated-weighted backward trajectory (CWT) analysis determines the originating source and transport of air parcels at the receptor location within a specific period (Seiber et al., 1994, Draxler et al., 1998). The HYSPLIT model (v4.1 Hybrid Single Particle Lagrangian Integrated Trajectory) was used to perform the CWT analysis (Draxler et al., 2018; Stein et al., 2015). The 72-hour back trajectories with a 3-hour time interval at 100 m of arrival height above the ground were calculated using monthly GDAS (Global Data Assimilation System) files (ftp:/arlf tp.arlhq.noaa.gov/pub/archives/gdas1) with a $1\circ \times 1\circ$ resolution. The estimated backward trajectories (BTs) were weighted with VOCs factors time series and averaged over 3-h intervals using a CWT model to locate air masses based on their concentrations. ZeFir (Petit et al., 2017), an IGOR-based interface, was used to construct the CWT graphs, as shown in supplementary Figure S6.

## 3.  Results and Discussions

### 3.1. NMVOCs concentration and temporal variation

The average daily concentrations of measured NMVOCs during the study period was 125.5 ± 37.5 ppbv. Figure 3 shows the daily time series and monthly mean concentrations of NMVOCs, inorganics and organics fractions of Nr-PM$_{2.5}$, O$_3$, NOx, SO$_2$, temperature, relative humidity, wind speed, and wind direction. Out of 173 detected NMVOCs, the level of three major species (Acetaldehyde, Acetone, and Acetic acid) were present 5-15 times higher than for other species, as shown in supplementary Figure S3. The monthly averaged concentrations of NMVOCs were higher during winter months from December (193.7 ppbv) to January (110.2 ppbv) till February (109.7 ppbv) than during the summer months, March (101.2 ppbv), April (137.8 ppbv) and May (150.8 ppbv). The averaged concentrations of NMVOCs (127±40 ppbv), as well as Nr-PM$_{2.5}$ (inorganics and organics) (102.8±51 $\mu g/m^3$), were higher during the winter months. The calm conditions and relatively lower planetary boundary layer during winters have slowed down the dispersion of the pollutants. In contrast, Nr-PM2.5 (39.8 ±20 $\mu g/m^3$) decreased drastically during the summer months, but NMVOC concentrations (122 ±32 ppbv) were similar to winters. This may be due to high temperatures during warmer periods may lead to more photooxidation of primary VOCs (Sahu et al., 2017), production of biogenic VOCs (Sahu et al., 2017; Baudic et al., 2016) and evaporation of volatile household products (Qin et al., 2021). While aerosol particles managed to disperse in the atmosphere due to the high planetary boundary layer and windy conditions. The difference in the emission sources' characteristics during both seasons may have also played an important role. During the winter, the PM2.5 exceeds most of days than the NAAQS standard. PM$_{2.5}$ exceeds standards more frequently than ozone. Approximately 80% of the days during the whole study period, PM$_{2.5}$ exceeds the NAAQS standards in the city, as shown in Figure 3.

The three most abundant NMVOCs were not considered in the PMF analysis, as explained in section 2.5. The remaining 170 NMVOCs, considered for the PMF analysis, varied from m/z 42. 034 to m/z 197.216. The average concentration of these 170 NMVOCs was 79.3 ±30.6 ppbv. The averaged concentrations during winters were 86.7 ±35 ppbv, a bit more than during summers at 68.3 ±17.2 ppbv. These NMVOCs belong to different families based on their chemical composition. They are categorised as aromatics (Ar_CxHy), simple non-aromatics (N_CxHy), furans (Furans), phenols (Phenols), oxygenates: first (CHO$_1$), second (CHO$_2$), and third order (CHO$_3$), nitrogen-containing compounds (CxHyNz and CxHyNzOn) and others. The others include high-order oxygenates (CHO$_4$) and some hydrocarbons (CxHy). The degree of unsaturation (i.e. the number of rings and/or double bonds) of more than 4 distinguishes aromatics (ArCxHy) from the CxHy family. This allowed us to identify important VOCs markers, their families, and their role in their atmospheric chemistry. Overall, during the study period, the highest contributing family belongs to oxygenates and aromaitcs. The CHO1, CHO2, and CHO3 families were 28.8% (~20.1 ppbv), 16.8% (11.7 ppbv), and 2% (1.4 ppbv) of total NMVOCs

concentrations. The contribution from Ar_CxHy, and N_CxHy were about 21.5% (~15 ppbv), and 10.6% (~7.4 ppbv), respectively. Nitrogen-containing compounds were relatively less present (5.6 % CxHyNz and 1.2% CxHyNzOn). 6.3% (~4.4 ppbv) and 3.7% (~2.6 ppbv) were contributed by Furans and Phenols at the site, the rest was included in others (3.4%). The CPCB notified the annual National Ambient Air Quality Standards (NAAQS) only for benzene as 5 μg/m$^3$ (~1.6 ppbv). While WHO recommended no safe level of exposure of benzene. The mean mixing ratio of benzene during the study period was found to be 2.9 ±1.9 ppbv which is around 2 times higher than the standard guidelines. Prolonged exposure or high short-term exposure to benzene adversely affects the health of citizens of the city due to its haematotoxic, genotoxic and carcinogenic properties.

All three abundant NMVOCs present at m/z 45.034 ($C_2H_5O$, acetaldehyde), 59.049 ($C_3H_7O$, acetone), and 61.028 ($C_2H_5O_2$, acetic acid) are oxygenated VOCs (OVOCs). The sources of these OVOCs could be direct emissions from biogenic and anthropogenic activities and the secondary/ photochemical processes. Diurnal variations of secondary formation, anthropogenic emission level, meteorological conditions, and PBL heights influence OVOCs/benzene ratios (Sahu et al., 2017; Tripathi et al., 2022; Sahu et al., 2016) to some extent. The diurnal patterns of acetaldehyde/benzene, acetone/benzene, and acetic acid/benzene ratios are plotted to check the influence of biogenic and secondary sources (see Figure 4 (a-c)). All the OVOCs/Benzene ratios are observed to increase during the daytime (10-18 h), similar to temperature variation. This infers the influence of these compounds' photochemical formation and/or biogenic emissions. The elevated OVOCs concentrations during the night confirm the influence of anthropogenic emissions.

Acetone and acetaldehyde are formed during photooxidation and ozonolysis of various terpenes and aromatics compounds emitted from multiple biogenic and anthropogenic sources (Lee et al., 2006a, b; Wang et al., 2020b). Acetone plays a major role in Ozone production. It can be transported to remote areas due to its long lifetime in the troposphere (~15 days) (Seco et al., 2007). The average concentrations of acetone during winters, late winters and summer were observed as 13.6 ±4.5 ppbv, 15.3 ±5.4 ppbv and 34.9 ±10.3 ppbv, respectively. The observed concentrations of acetone in Lucknow are on the higher side of the range of measured concentrations in other Indian cities. The reported average concentrations of acetone in the present study are comparable to Delhi (whole year) is ~16.7 ppbv (13-15 ppbv during winters) (Jain et al., 2022) but higher than Ahmedabad at 5.35 ±1 ppbv during late winters (Sahu et al., 2016), and Mohali as 5.9 ±3.7 ppbv during summers(Sinha et al., 2014). This shows the presence of more OVOCs in cities within the IGB region than in other cities of India.

Figures 4 (a) and 3 (b) show that the diurnal variation of acetaldehyde and acetone, respectively, starts increasing from 9 h in the morning to 16h in the evening. The acetaldehyde and acetone had their morning maxima at around 10:00 LT and 11:00 LT, respectively, and later during the morning rush hours of vehicular emissions (8-10h). This trend is similar to a previous study in Ahmedabad (Sahu and Saxena, 2015). It indicates the secondary formation of acetone and acetaldehyde from terpenes and aromatics emitted from vehicles. The OH reaction rate constant of the hydroxyl radical with acetaldehyde ($15 \times 10^{-12}$ cm$^3$ molecule$^{-1}$ s$^{-1}$) is significantly higher than the reaction rate constant of the hydroxyl radical with acetone ($0.17 \times 10^{-12}$ cm$^3$ molecule$^{-1}$ s$^{-1}$), indicating faster degradation of acetaldehyde than acetone (Atkinson and Arey, 2003). Figure 4 (c) shows the diurnal variation of acetic acid and acetic acid/ benzene ratio. Acetic acid is one of the most abundant VOC species in the atmosphere globally, having a half-life of more than one day. It contributes to atmospheric acidity (Chebbi and Charlie, 1996) and is responsible for 30% acidity of the wet deposition in polluted urban areas ( Seco et al., 2007, Pena et al., 2002). This compound also has toxic effects on human health. Residential wood combustion is one of the critical sources of acetic acid (Bruns et al., 2017). High levels of acetic acid have also been reported from aged open biomass (hay and straw) burning plumes (Brilli et al., 2014), a variety of biomass fuel (Stockwell et al., 2015), and natural gas (Gilman et al., 2013). The average concentration of acetic acid during the whole study period is about 10.3 ±4.1 ppbv, highest during winters at 15.2 ±3.5 ppbv and lower during summers at 6.9 ±2.1 ppbv. The observed increased concentrations of acetic acid in the winter and lower concentration in the summer may demonstrate its production influenced by open biomass burning crops in nearby fields and residential wood combustion for heating and cooking purposes in Lucknow. The diurnal pattern of acetic acid concentrations shows high concentrations during the night, which infers its accumulation.

### 3.2. Characteristics of selected PMF factors

This section includes a discussion of the selection of the source apportionment solution and its interpretation. The NMVOCs factors are identified based on their mass spectra, diurnal and temporal variation, and correlation with external tracers. For the first time, we have included mass spectra of 170 NMVOCs from m/z 42.034 to m/z 197.216 in the PMF analysis. The three abundant NMVOCs (compounds below m/z 42) detected by PTR-ToF-

MS, acetaldehyde, acetone, and acetic acid, are not included in PMF analysis. Including these NMVOCs in the PMF analysis resulted in biased solutions where only these ions are well explained. Additionally, a few small alkanes and alkenes (C1-C4) compounds, which are not detected by PTR-ToF-MS, are excluded from PMF analysis. However, previous studies have found that these ions are minor contributors to SOA formation. Included compounds (above m/z 42) are major contributors to SOA formation and dominant markers of various sources. As explained in section 2.4, the optimum solution after the PMF analysis chosen is a 5-factor solution. This selected 5-factor PMF solution exhibits distinct mass spectral characteristics related to different sources and atmospheric processes. Figure 5 shows the intricate plots of the profile and diurnal variation of the 5-factor solution. The five factors are Traffic, SFC 1 (solid fuel combustion), SVOC (secondary volatile organic compounds), SFC 2, and VCPs (volatile chemical products) after thoroughly investigating markers, chemical species and their families, diurnal variation, and relation to meteorological parameters and external measurements. The diurnal variation of the factors for two seasons (winter and summer) were compared, as shown in supplementary Figure S7. The timeseries of the five factors resolved from NMVOCs mass spectra are co-related with external measurements such as oxygenated organic aerosols (OOA), Black carbon (BC) concentrations, CAAQMS data (WD, WS, RH, Temp, NO, NO2, NO$_X$, SO$_2$, O$_3$) as given in Figure 6.

### 3.2.1.    Factor 1: Traffic

The first factor is identified as traffic. It is characterized by the presence of aromatics, such as benzene (m/z 79.053, C$_6$H$_6$H+), toluene (m/z 93.07, C$_7$H$_8$H+), xylene (107.09, C$_8$H$_{10}$H+), C9-aromatics (121.1, C$_9$H$_{10}$H+), and C10-aromatics (135.12, C$_{10}$H$_{14}$H+). 56% of the total aromatics are explained by this factor, as shown in Figure 7. The explained variation of individual NMVOCs, such as C$_6$H$_6$H+, C$_7$H$_8$H+, and C$_8$H$_{10}$H+, by the traffic factor, is around 0.56, 0.77, and 0.76, respectively, as shown in Figure 8 (a). The NMVOC's traffic factor shows a temporal correlation (Pearson r$^2$ ~ 0.74) with nitrogen oxides (NO$_X$), which is also an indicator of vehicular emissions (Figure 6). Also, this factor has a good correlation (Pearson r$^2$ ~ 0.65) with the AMS_HOA (PMF-resolved factor from HR-ToF-AMS), as shown in Figures 9 (a) and 9 (b). This AMS_HOA factor is characterized(Lalchandani et al., 2021)NMVOCs, NOx and primary OA. These NMVOCs and primary OA also exhibit similar temporal and diurnal variation, having sharp peaks during morning and evening hours, as shown in supplementary Figure S8. This diurnal pattern indicates the vehicular commute pattern in the city, and the high density of vehicles on the roads during rush hours in the morning and evening. The diurnal pattern is compared between two seasons, winters and summers, and also shows a similar pattern in supplementary Figure S7. The traffic factor in previous studies observed similar markers and diurnal patterns in Delhi (Jain et al., 2022; Wang et al., 2020a) and Beijing (Wang et al., 2021b), indicating a similar commute pattern in most of the urban cities. Other source-specific studies also identified similar markers for vehicular emissions (Cao et al., 2016; Caplain et al., 2006). The back trajectory analysis of the factor (CWT graph), given in Supplementary Figure S6, shows the probable sources of traffic present near the sampling site.

### 3.2.2.    Factor 2: Solid Fuel Combustion (SFC 1)

Another factor which is resolved is Solid Fuel Combustion (SFC 1) has the highest contribution from furans and substituted furans (~36%) and nitrogen-containing compounds (34%), as shown in Figure 7. The prominent signals of acrylonitrile (m/z 54.034, C$_3$H$_4$N), furan (m/z 69.033, C$_4$H$_5$O), pyridine (80.054, C$_5$H$_6$N) furfurals 81.036, C$_5$H$_5$O), furaldehyde (m/z 97.027, C$_5$H$_5$O$_2$), dimethyl furan (97.064, C$_6$H$_9$O), and C$_3$H$_3$N$_2$O$_3$ (115.012) also contribute to the factor's mass spectra as shown in Figure 5 (a). This factor profile is characterized by the strong peak of acetonitrile (m/z 42.034, C$_2$H$_4$N) with an explained variation of about ~0.49, as shown in Figure 5 (b). Acetonitrile is considered a unique marker of biomass burning (Holzinger et al., 1999). Furans and nitrogen-containing compounds are mostly emitted from combustion processes (Coggon et al., 2019), cooking fires, burning of peat, crop residue and biomass fuel such as wood, and grasses (Stockwell et al., 2015). Studies have also shown that furans and nitrogen-containing compounds have a high potential to form secondary organic aerosols and particles.   Other markers, nitrophenol (m/z 140.033, C$_6$H$_6$NO$_3$) and methoxy nitrophenol (m/z 154.054, C$_7$H$_8$NO$_3$) explained by SFC 1 factor profile of ~0.53 and 0.52, respectively. It is reported that phenols in a biomass smoke plume react with NOx to form nitrophenol, considered a unique marker for aged biomass burning smoke (Harrison et al., 2005; Mohr et al., 2013). Nitrophenols and other nitrogen-containing aerosols act as cloud condensation nuclei (Kerminen et al., 2005; Laaksonen et al., 2005; Sotiropoulou et al., 2006), and contribute to the formation of SOA and light-absorbing brown carbon aerosols (Mohr et al., 2013; Laskin et al., 2009). SFC 1 factor correlates with organics fraction of Nr- PM2.5 (Org_Hr), NO$_3$_Hr (inorganics NO3 of Nr-PM$_{2.5}$) and RH well with Pearson r$^2$ ~ 0.46, 0.53, and 0.47, respectively. (Lalchandani et al., 2021; Shukla et al.,

2021), as given in Figure 6 and Figure 9 (d). Thus, we interpret SFC 1 is more related to conventional biomass
burning at the site. The diurnal pattern of the SFC 1 from NMVOCs (Figure 5) shows peaks during cooking times,
morning (7:00-8:00) and evening (19:00-21:00). The domestic usage of biomass for cooking and other purposes
is one of the leading factors for primary emissions of gas-phase (SFC 1) and particle-phase oxygenates (OOA).
The city is surrounded by various agricultural fields, which generally involve open biomass-burning activities.
The back trajectory analysis of the factor also shows the probable sources in nearby areas, mainly coming from
the west direction of the sampling site (supplementary Figure S3). This argues that this factor is also influenced
by the aged biomass-burning plume, transported from sources located on the outskirts of the city and nearby
districts.

### 3.2.3.  Factor 3: Solid Fuel Combustion (SFC 2)

The third factor, Solid Fuel Combustion (SFC 2), was identified in the 5-factor solution. This component is
basically only present in December. The factor's mass spectra are characterized by peak signals of methyl furan
(m/z 83.049, $C_5H_7O$), phenol (m/z 95.049, $C_6H_7O$), cresol (m/z 109.06, $C_7H_9O$), catechol (m/z 111.043, $C_6H_7O_2$),
phenyl butanedione (m/z 163.115, $C_{10}H_{11}O_2$), hexene (m/z 85.093, $C_6H_{13}$), as shown in Figure 5 (a). The SFC 1
and SFC 2 factor profile is compared with each other in Figure 8 (a). It explains the similar NMVOCs are present
in the factors, but the intensity of the signal is different. This is due to the difference in the emission sources and
chemical pathways of formation. Lower ambient temperature and high relative humidity during this month are
responsible for the different chemical pathways for the fate of compounds. For example, High molecular weighted
and more substituted phenolic compounds such as guaiacol (m/z 125.059, $C_7H_9O_2$) and cresol are released at the
early stages of the smouldering stage of the fire (lower temperature), and low molecular weighted phenols are
released during later stages (high temperature) (Stewart et al., 2021b). The higher explained variation from cresol
(~0.8) and guaiacol (0.21) to the factor's profile indicate their new emissions from residential heating activities
and the burning of sawdust (Stewart et al., 2021b), as shown in Figure 8 (b). Other compounds like phenols (0.27)
and hexene (~0.62) are explained by this SFC 2 factor's profile. These two compounds are being reported in the
emissions from local biomass burning of wood in an Indian city (Delhi)(Stewart et al., 2021c). The factor profile
explains 53% of phenols, 23% of second order oxygenates, 30% of furans and 21% of nitrogen-containing
compounds. Commonly used domestic fuels other than liquid petroleum gas (LPG) in the Indian sub-continent
are cow dung, fuelwood, and peat, in different proportions depending upon their composition and availability. A
previous study (Stewart et al., 2021a) observed phenols are released from the combustion of fuelwood (22-80%),
followed by crop residue (32-57%), cow dung cake (32-36%), and municipal solid waste (24-37%). The
combustion process at a higher temperature leads to the depolymerisation of lignin content in the biomass, which
allows the aromatisation process to give off phenols, substituted phenolic compounds, and non-substituted
aromatics (Sekimoto et al., 2018; Simonelt et al., 1993). The lower ambient temperature during December is also
responsible for increased burning activities for cooking and heating purposes. The diurnal variation of SFC 2
shows its prominence during evening hours and accumulation during the late evening (21:00) till mid night. The
correlation coefficient between SFC 2 and black carbon concentrations is ~0.4. The factor SFC 2, derived from
the VOC mass spectra, is less related (Pearson $r^2$ ~0.38) to the AMS_MO-OOA as shown in Figure 9 (c) and
supplementary Figure S8. AMS_MO-OOA is characterised by higher m/z 44 ($CO_2$) and m/z 43 ($C_2H_3O$) fractions
than the primary OA sources. This factor is comparatively more oxidised, having an O/C ratio of ~0.89 than
AMS_LO-OOA (O/C ratio ~0.62). It may be interpreted that SFC 2 is influenced by fresh oxidation of primary
biomass burning emissions. Moreover, as shown in supplementary Figure S3, the CWT plots show no evidence
of its long-range transport.

### 3.2.4.  Factor 4: Secondary volatile organic compounds (SVOC)

The fourth factor, secondary volatile organic compounds (SVOC), has the highest contribution from second-order
oxygenates (40 %) and third-order oxygenates (40%), as shown in Figure 7. The relative composition of the profile
of the factor reveals significant signals of acetic acid (m/z 77.019, $C_2H_5O_3$), propylene glycol (m/z 77.048,
$C_3H_9O_2$), methylglyoxal (m/z 73.028, $C_3H_5O_2$), methyl methacrylate (93.033 $C_6H_5O$), and $C_5H_9O_2$ (m/z 101.059),
given in Figure 5 (a). Lower contributions from first-order oxygenate than the second, and third-order oxygenates
indicate that these OVOCs are products of various photochemical and oxidation processes in the atmosphere
instead of their direct emissions. The diurnal mean concentration of the SVOC factor in Figure 5 (b) shows distinct
day-to-night variation, following the pattern of solar radiation. The mean concentration increases during the
morning (8:00), peaks during the afternoon hours (12:00-15:00) and decreases towards the evening (20:00). The
nighttime concentration of the factor is lowest due to the absence of photochemical activity at night. Small organic
acids like formic acid (m/z 47.012, $CH_3O_2$) could potentially come from the photooxidation of furans and
aromatics (Wang et al., 2020b; Stewart et al., 2021b), which contribute 42.2% to the SVOC factor's profile (Figure
7). Other compounds like methoxyphenols are released by biomass burning, which is further photo-oxidised,
resulting in the formation of SOA (Yee et al., 2013; Li et al., 2014). Figure 8 (b) shows the explained variation of
these compounds, such as vanillin (methoxyphenol) and syringol (2,6-dimethoxyphenol) to the SVOC factor is
~0.57 and 0.41, respectively, relatively high. This also confirms the association of products and intermediate
products of photochemical reactions with the SVOC factor. The temporal variation of this factor has no significant
correlation with any of the AMS factors or external tracers.

### 3.2.5. Volatile chemical products (VCPs)

The Volatile Chemical products (VCPs) factor is identified with prominent signals of ethanol (m/z 47.049, $C_2H_6O$)
and naphthalene (m/z 129.05, $C_{10}H_9$), given in Figure 5(a). 76.6% of the ethanol contributes to the VCPs factor
(Figure 7). Volatile chemical products show good temporal co-relation with a solvent-based NMVOCs species,
Acetone, with Pearson $r^2 \sim 0.6$ (Figure 6). Ethanol is used as a solvent in the paint, solvent-based, textile, plastics,
and automobile industries. Many such kinds of industries (solvent-based and textile) are present in the close
vicinity of the sampling site, possibly due to the high concentration of formaldehyde and ethanol. Shorter-life
spans of ethanol (~3-4 hours) in the atmosphere confirm its emissions from local sources instead of transport from
regional sources. The relative contribution of naphthalene is about 28.3%, respectively, to the factor. Other
dominant signals of naphthalene diamine (m/z 159.102, $C_{10}H_{11}N_2$) and methoxy benzopyranone (m/z 177.056,
$C_{10}H_9O_3$) relatively contribute about 34.5% and 44.45% to the factor. Naphthalene is present in ambient air due
to emissions from the industries such as metal industries, chemical manufacturing industries, and pharmaceuticals
(Preuss et al., 2003). Naphthalene is also used as an intermediate product in coal tar, dyes or inks, leather tanning
and asphalt industries (Jia and Batterman, 2010). It is classified as a possible human carcinogen and precursor of
atmospheric SOA (Tang et al., 2020; Jia and Batterman, 2010). There are very sharp peaks in the concentrations
of ethanol, naphthalene, naphthalene diamine and benzopyrene in the high-resolution timeseries, as shown in
supplementary Figure S9. This may be due to the influence of particular activity in near-by industries. A
conglomerate of the industries is present in the southwest direction of the sampling site within and outside the
city, as shown in Figure 1. The direction of the wind changes to the southwest during summers may have brought
the high levels of naphthalene and its derivatives emitted from these industrial areas to the sampling site. The
CWT graph also shows the strong influence of the source present in the southwest direction of the sampling site
(supplementary Figure S3). A previous study has found that among the emitted OVOCs from sewage sludge, first-
order OVOCs constituent ~60%, followed by high-order OVOCs (Haider et al., 2022). Interestingly, there are
three sewage treatment plants located near the sampling site. They may have also influenced the concentrations
of OVOCs at the sampling site. The influence of factor contribution during summertime is probably due to the
increased production of naphthalene, and ethanol from their local industrial sources and secondary formations at
higher temperatures, as shown in the time series of the factors (supplementary Figure S8).

### 3.3. OFP and SOA yield from individual sources

Based on the method explained in section 2.5, the ozone formation potential was calculated for each factor after
considering the MIR values of NMVOCs species as given in supplementary Table S2. The relative contribution
of each NMVOCs to the individual factor after PMF analysis is multiplied by the averaged individual
concentration of the NMVOC species. The highest contributor species to the ozone formation potential is toluene,
followed by xylene, isoprene, and methyl cyclohexene. The distribution of individual sources to OFP is shown in
Figure 10 (a). Toluene, xylene, and isoprene were found to be the highest contributor in terms of OFP in other
Asian cities, including Guangzhou and Beijing (Zheng et al., 2009; Zhu et al., 2016; Zhan et al., 2021; Duan et
al., 2008). In the previous study in Delhi, it has also been noticed that the contributions of aromatics (xylene and
toluene) have a substantial effect on the ozone formation potential (Tripathi et al., 2022). The traffic factor
contributes maximum to the OFP among all the factors with 34.6%, followed by SFC1 (23.9%), then SFC
2(14.5%), SVOC (13.5%) and VCPs (13.5%). Similarly, the contribution towards the formation of SOA is also
estimated for each factor with the SOA yield of individual NMVOCs, as given in supplementary Table S3. The
overall SOA yield is influenced by toluene, benzene, phenol, naphthalene, xylene, methyl furan, and trimethyl
benzene. These compounds mostly belong to the aromatics, and first-order oxygenates family. The measured SOA
from HR-ToF-AMS may be considered as the sum of a more-oxidised oxygenated OA factor (AMS_MO-OOA)
and one low-oxidised oxygenated OA (AMS_LO-OOA) factor (Lalchandani et al., 2021). The five highest
contributors to SOA formation potential were correlated with the measured SOA in the supplementary figure S10,
the high-resolution time series shows the co-occurrence of high and low peaks of benzene, toluene, and xylene
with measured SOA during the day and night hours. This shows the significant role of aromatic NMVOC species

in the formation of SOA. The primary factors, Traffic and SFC 1, are the highest contributors to the SOA formation, with 28% and 27%, respectively, as shown in Figure 10 (b). These factors are ridden with the highest SOA formation contributing NMVOCs species. Previous studies have also found that aromatic hydrocarbons contributed more than 95% to the SOA formation potential in other Asian cities (Qin et al., 2021a; Zhan et al., 2021). It was observed that the sources related to vehicular emissions (diesel and petrol-driven vehicles), paddy stubble fire, and garbage fire emissions were the most contributing factors to ozone formation potential in Mohali (Kumar et al., 2020). In the present study, the SVOC factor contributes 22% to the SOA formation, with maximum contribution from high-molecular oxygenated species. The SFC 2 and VCPs are contributing less towards the SOA formation.

In contrast, this sequence is not similar to the relative contribution of the sources according to their concentration (Figure 10 (c)). The source contributing to the highest concentration of NMVOCs is SFC 1, followed by traffic, SVOC, and VCPs. The lowest contributor is SFC 2. This comparison shows the importance of the source of NMVOCs towards SOA and ozone chemistry. The factor contributing the highest to the concentration of NMVOCs may not necessarily influence the ozone and SOA formation similarly. These values estimate the potential for ozone and SOA formation and do not indicate the actual yields of ozone and SOA. This estimation method represents the complex behavior of NMVOCs, NOx and solar radiation for producing tropospheric ozone and SOA. There are many NMVOCs species with unknown ozone and SOA yield values. One needs to understand the chemical fates and pathways of many NMVOCs by mimicking real-time atmosphere in smog-chamber studies or through computational modelling studies. More research on this section is needed. Nonetheless, other parameters, including solar radiation and concentration of oxides of nitrogen, also play a key role in the formation of ozone in the troposphere. In reality, OFP and SOA do not provide complete information about how VOCs influence O3 and organic aerosol chemistry zone formation in Lucknow is more sensitive to NMVOCs concentrations than NOx, similar to other Asian cities. So, Decreasing the VOCs/NOx ratio would also help reduce the secondary pollutants (O3 and SOA). It is observed that vehicular emissions were the main source of aromatics (benzene, toluene, xylene). Therefore, vehicular emission control strategies should be implemented to reduce aromatic (BTEX). Stringent implementation of policies and fuel-efficient standards related to vehicular emissions in Japan and South Korea have primarily improved the air quality (13-17% reduction in NMVOCs) (Wang et al., 2014). In the present study, one of the key observations was that toluene is the main contributor to SOA and ozone production potential. This illustrates that targeting other sources of some NMVOCs (toluene and xylene) will also enhance its control. For example, paint solvents (source of ethylbenzene and xylene) and printing products (source of toluene) were targeted in a city, Hong Kong, where the VOC content of 172 types of consumer products was prescribed by the respective government (Lyu et al., 2017). In the present study, other potential contributor species are methyl cyclohexene (for ozone) and naphthalene (for SOA). These compounds are related to volatile chemical products, as found in the PMF analysis in Lucknow. This infers stringent policies related to solvent-based industries such as textile, automobile, paints, and disinfectants are needed. Regulation and control of NMVOCs content in manufacturing and use of solvent-based products such as pants, disinfectants, fungicides, and insecticides should also be implemented. In China, various industries implemented end-of-pipe measures to control NMVOCs, such as refineries, plant oil extraction, gasoline storage and service stations, pharmacies, and crude oil storage and distribution (Wang et al., 2014) It is also estimated that China's end-of-pipe technologies and new energy-saving policies would help decrease about one-third of NMVOC emissions (Zhang et al., 2020). Phenols and Furans were observed as one of the highest contributors to SOA formation potential related to solid fuel combustion. This suggests controlling solid fuel usage for residential energy and crop-residue burning in the fields within and around the city Lucknow. Firewood burning during the heating period and domestic in-fields straw burning have substantially reduced emissions from biomass burning in China (Wu et al., 2020) . (Derwent et al., 2007) reported that reactivity-based VOC control measures might be more effective than mass-based regulations in controlling ozone and secondary organic aerosol formation. The present study also suggests that the reduction of VOC, especially from vehicular emission is needed for the abatement of O₃ and SOA formation in urban areas.

## 4. Comparison with other Indian and Asian cities

Figure 11 represents mapped pie charts to compare overall NMVOCs concentrations and relative source contributions in different Asian and Indian cities. The earlier studies reported the total NMVOCs concentrations between 15-35 ppbv in different cities in China during winter (Hui et al., 2018; Wang et al., 2021a; Yang et al., 2018; Wang et al., 2016). The highest concentration of NMVOCs was found in Wuhan city (~34.6 ppbv), with maximum contributions from alkanes and oxygenated VOCs(Hui et al., 2018). The relative composition of

sources of NMVOCs found in Wuhan was Industrial/Solvent usage (29.9%), followed by traffic (24.4%), fuel evaporation (23.87%), biomass burning (19.3%) and biogenic (2.5%). The urban site in Beijing reported the maximum contribution from secondary VOCs (54.6%), followed by biomass burning (24.4%) and traffic (21%)(Wang et al., 2021a), while the rural site in Beijing had significant contributions from biomass burning (37%)(Yang et al., 2018). Industrial and Traffic contributed similarly at the rural site in Beijing (~31.5%). The difference in source profiles and contributions in urban and rural areas inferred the need for different control strategies and policies in the country (Zhang et al., 2020). It is found that vehicular emissions and biomass burning sources contribute to NMVOCs concentrations (average ~21.5 ppbv) overall 50%, and 41%, respectively, during summers, in a land-locked urban city, Lhasa, Tibet(Guo et al., 2022) while Industrial/Solvent usage contributed 68% to NMVOCs (average ~33.7 ppbv) in Tokyo, Japan(Fukusaki et al., 2021b). It is interesting to note that near the coastal region in Hong Kong, 63.7% and 13.5 % of NMVOCs contributions (average ~9.8 ppbv) are related to biomass burning and ship emissions (Tan et al., 2021)various air pollution control strategies implemented for over a decade, NMVOCs and $O_3$ concentrations did not decrease significantly in Hong Kong (Lyu et al., 2017)A previous study in Kathmandu(Sarkar et al., 2017), Nepal, demonstrated that biomass co-fired brick kilns (29%) and traffic (28%) contributes to SOA production significantly. Other sources, such as Industrial/ Solvent-usage, biomass burning, and biogenic-related emissions, dominate the city.

Earlier source apportionment studies over the NMVOCs mass spectra conducted in Indian cities are limited to two cities in the upper IGB region, Delhi (full year) and Mohali (summer). Comparing the urban and sub-urban sites of Delhi found that vehicular emissions are dominant at both sites, with relatively less contributions to NMVOCs in the sub-urban region (36%) compared to the urban region (57%). Throughout the year, traffic emissions dominated the NMVOCs concentration (31%), with comparable contributions from biomass burning (28%) and secondary formations (31%) overall in Delhi. Mohali is located upwind of Delhi city, with maximum contributions from biomass burning (47%), followed by traffic (25%) and secondary formations (16%). The industrial source contributed about 5% and 12% to NMVOCs concentrations in Delhi and Mohali, respectively. While in the present study, it is found that the solid fuel combustion-related emissions majorly (41.3%) contributed to NMVOCs concentrations in Lucknow, located in the central IGB region. The traffic-related emissions (23.5%) and secondary formations (18.6%) are relatively less contributing to NMVOCs as compared to upper IGB region cities (Delhi and Mohali). Moreover, the volatile chemical products emitted more during the summer period in Lucknow than compared to Delhi and Mohali. Solid fuel combustion sources aided concentrations of NMVOCs in both Mohali and Lucknow significantly. This may be due to both cities being located downwind of the widespread area of agricultural fields. Both cities observed relatively less formations of secondary volatile organic compounds, suggesting the dominance of fresh emissions over aged compounds in the air mass. Overall, the ambient concentrations of NMVOCs in Indian cities are majorly influenced by solid fuel combustion emissions, vehicular-related emissions, secondary formations, and industrial sources. This suggests the need for control measures and policies implemented for specific sources country-wide and specific to the city.

## 5. Conclusion

This study investigated the high time-resolved chemical characterisation of NMVOCs in Lucknow between December 2020 and May 2021. The mass spectra of the NMVOCs were used to perform source apportionment and study the diurnal variations. The individual species were identified as per their chemical formula and exhibited large temporal fluctuations. The highest NMVOCs concentrations during winters were due to their increased emissions from solid fuel combustion and stagnant conditions due to less mixing height. The warmer period between April and May showed the influence of high photochemical activity and regional transport. The major industries are observed in the southwest direction of the sampling site, which may be responsible for highly volatile chemical products during summer. The five major factors resolved from source apportionment were a traffic factor, two solid fuel combustion factors, secondary VOCs, and VCPs. The primary sources, such as traffic factor and solid fuel combustion, exhibited a stronger correlation with organic aerosol resolved factors, indicating their expected time of origin from similar sources. The traffic factor had a similar profile found in Delhi, which suggested a similar vehicular pattern and fuel composition in different urban centers of the IGB region of India. The biomass burning factors in Lucknow had distinct profiles from Delhi due to different cooking or domestic fuel consumption and cropping patterns. Moreover, the regional transport of secondary volatile organic compounds was also observed in the back-trajectory analysis. The primary first-order oxygenates most contributed to the VCPs factor, while the secondary VOCs factor had contributions from second and third order oxygenates. The highest contributing factor towards the NMVOCs emissions in Lucknow was solid fuel combustion (SFC 1) and traffic. The PTR-ToF-MS resolved source factors of

NMVOCs were correlated with HR-ToF-AMS resolved factors, Nr-PM$_{2.5}$ (organics and inorganics), and supporting measurements (BC, NOx, SO$_2$, O3) to analyze their common sources and diurnal patterns. The Ozone and SOA formation potential from individual NMVOCs species and sources were also estimated using MIR and SOA yield values-based methods, respectively. There is a scope for improving these estimates as these values represented the potential for the formation of SOA and O3, not the actual yields. It is found that a few of the NMVOCs species are significantly responsible for secondary pollutant formations. Stringent policies and control actions regarding aromatics (benzene, toluene, xylene, and naphthalene) and oxygenates (phenol and furans) could reduce the NMVOCs emissions drastically. The sources potentially contributing to SOA and ozone formations are traffic, SFC and VCPs. Further control measures and end-to-pipe technologies to reduce emissions from solvent-based industries, consumer products, residential and domestic biomass burning, and vehicular fleets are required to mitigate the health and environmental impacts of NMVOCs and secondary pollutants. The results of this study suggest that to refine the strategies to improve air quality in urban regions of India, particularly the Indo-Gangetic Plain, comprehensive measurements of VOCs are necessary to characterize their emission sources and understand their photochemical processes. This work highlights those local emissions, meteorology, city planning and implementation of the policies in the IGB region highly influence the NMVOCs sources. Further studies focusing on VOCs-secondary organic aerosol interactions would help identify the gas-particle partitioning, ageing and transport of pollution in the region.

## 6. Data availability

The data is available on the request with the corresponding author.

## 7. Author Contribution

**Vaishali Jain:** Conceptualization, data curation, Methodology, Software, Validation, Formal analysis, Investigation, Writing– original draft, Writing– review & editing, Visualization, **Nidhi Tripathi:** Investigation, Data curation, Validation, Writing– review & editing, **Sachchida N. Tripathi:** Conceptualization, Writing– review & editing, Supervision, Project administration, Funding acquisition **Mansi Gupta:** Investigation, Data curation, Validation, Writing– review & editing, **Lokesh K. Sahu:** Resources, Methodology, Validation, Writing– review & editing **Sreenivas Gaddamidi:** Investigation, Data curation, Validation, **Ashutosh K. Shukla:** Validation, Writing– review & editing, **Vishnu Murari:** Formal analysis, Validation, Writing– review & editing, **Andre S.H. Prevot:** Methodology, Validation, Writing– review & editing.

## 8. Competing interests

The authors declare that they have no known competing financial interests or personal relationships that could have appeared to influence the work reported in this paper.

## 9. Acknowledgements

LKS, NT and MG acknowledge Prof. Anil Bhardwaj, Director, Physical Research Laboratory (PRL), Ahmedabad, India, for the support and permission to deploy PTR-TOF-MS during the experimental campaign. SNT and VJ gratefully acknowledge the financial support provided by the Swiss Agency for Development and Cooperation, Switzerland, to conduct this research under project no. 7F-10093. 01. 04 (contract no. 81062452). SNT also acknowledges the support from Duke University, Office of Research Support, Subaward no. 349-0685. The authors would like to acknowledge the support from UPPCB (Uttar Pradesh Pollution Control Board) for the set-up of the campaign site. The authors would also like to acknowledge the support of PSI and Centre of Excellence (ATMAN) approved by the office of the Principal Scientific Officer to the Government of India. The CoE is supported by philanthropies including Bloomberg Philanthropies, the Children's Investment Fund Foundation (CIFF), the Open Philanthropy and the Clean Air Fund.

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

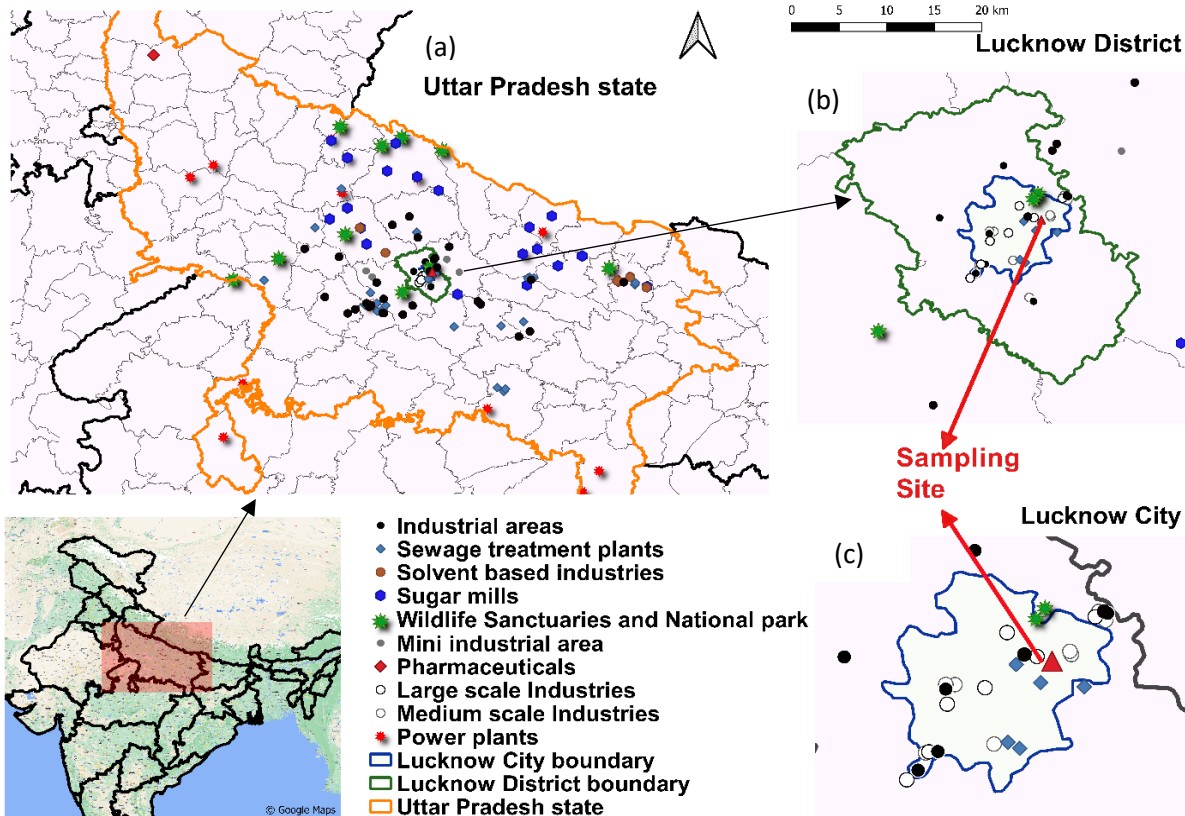

**Figure 1: Detailed map of (a) Uttar Pradesh, (b) Lucknow district and (c) Lucknow City with highlighted sampling site and major potential point-sources of NMVOCs**


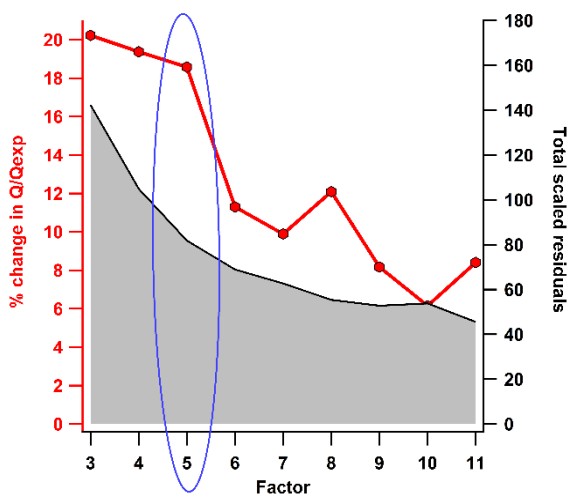

**Figure 2: The Q/Qexp plot (% change) and total summed scaled residuals for each factor solution**




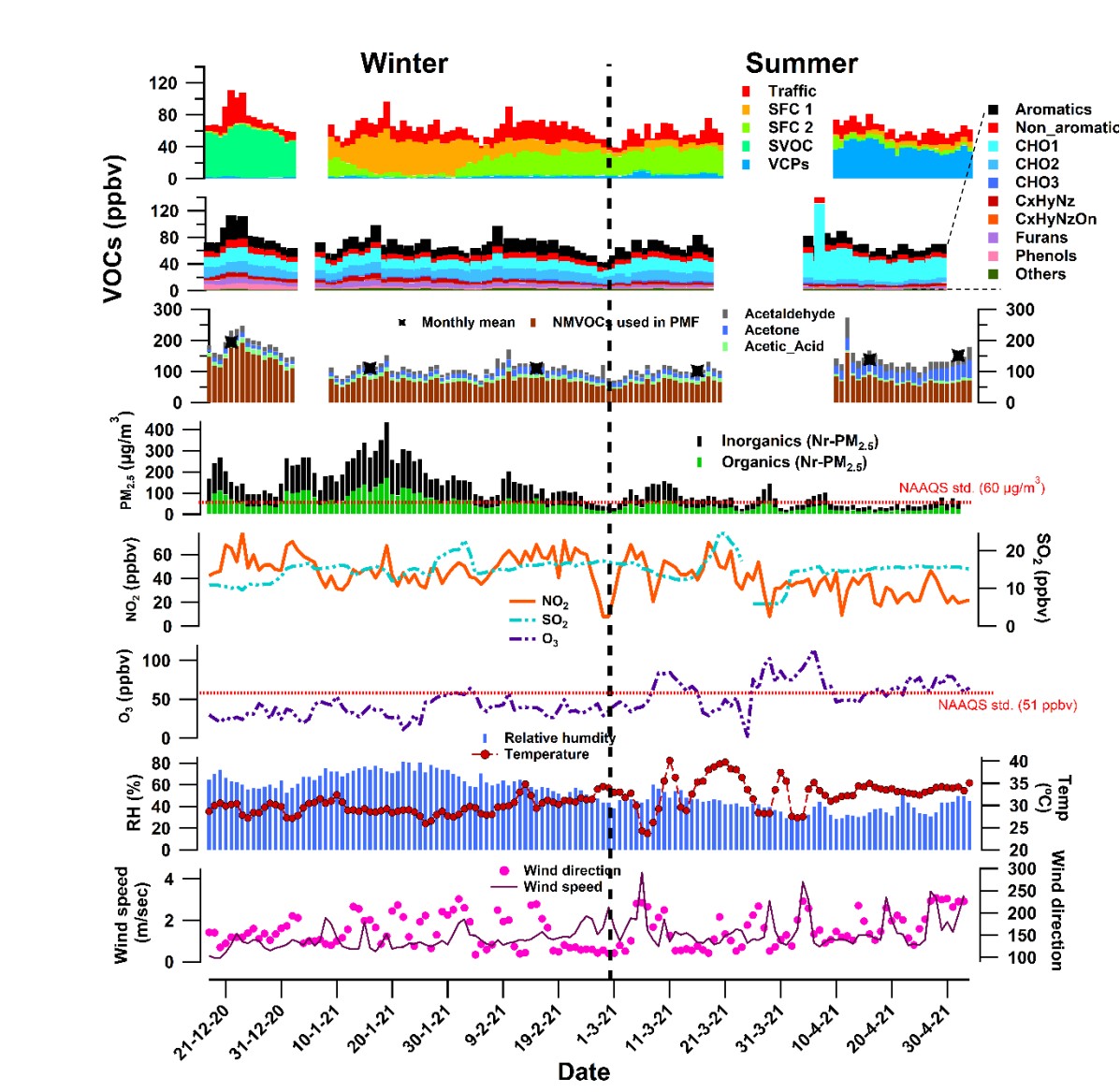

**Figure 3: Daily averaged time series of acetaldehyde, acetone, and acetic acid, other NMVOCs, PM₂.₅ and its organic fraction, NO₂, SO₂, O₃, temperature, relative humidity, and wind speed and direction**

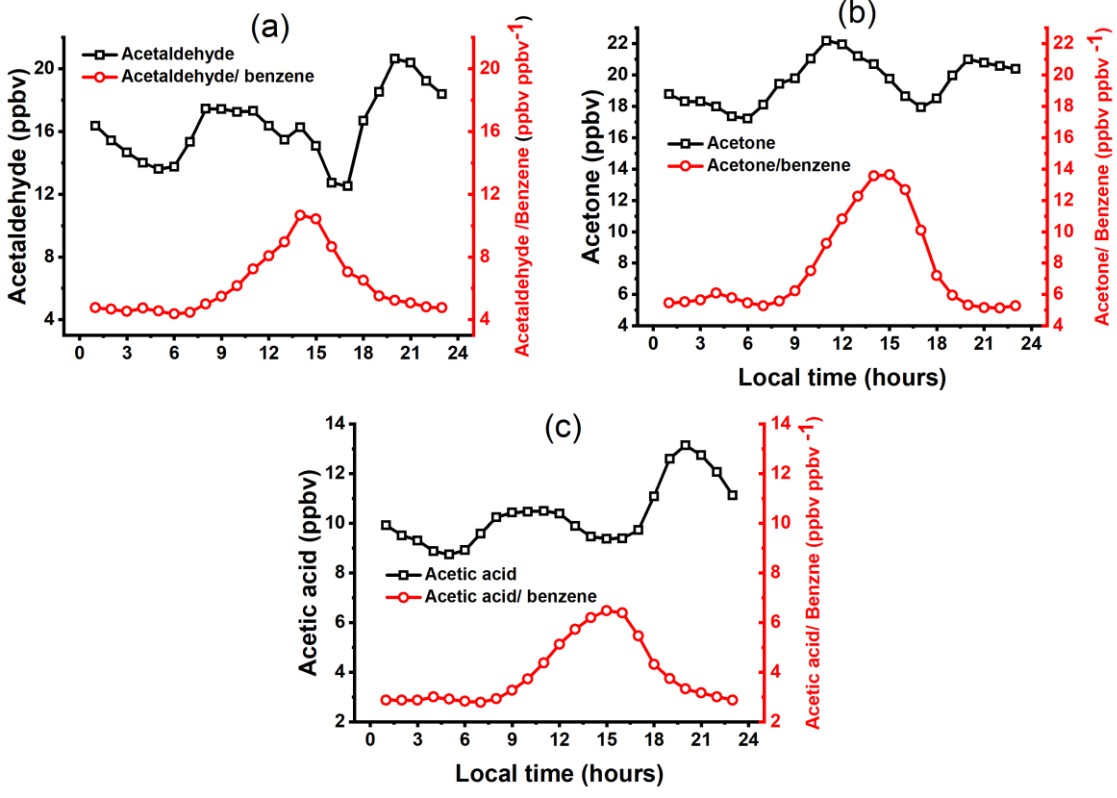

**Figure 4: Diurnal variations over the whole study period for (a) Acetaldehyde and Acetldehyde/benzene ratio, (b) Acetone and Acetone/benzene ratio, and (c) Acetic acid and Acetic acid/benzene ratio**
















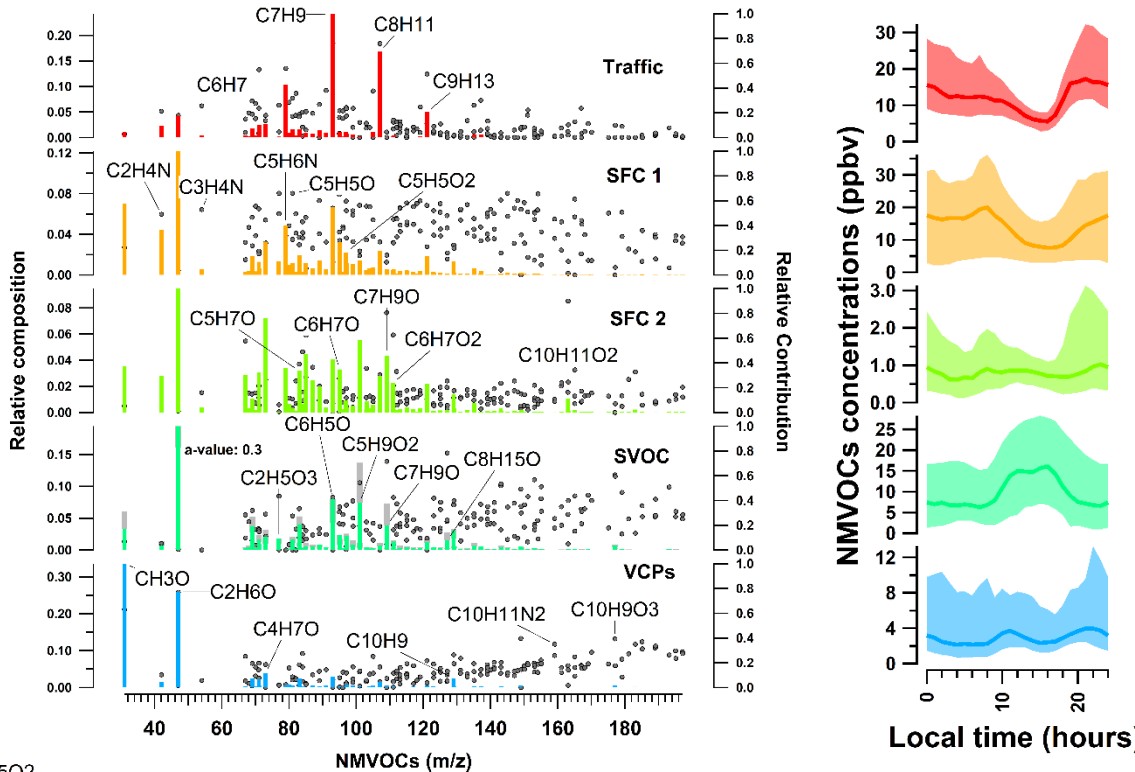

**Figure 5: Profile and diurnal variation of individual factors of selected 5-factor solution after PMF analysis at Lucknow for the whole study period. In (a), the left axis represents the relative composition of each factor, given by the vertical bars. The sum of all the bars at different m/z for each factor is 1, and the right axis represents the relative contribution of each factor to a given m/z, shown as grey dots. The grey bars in the SVOC factor represents the degree of constraint on the known source profile and time series. In (b) the middle dark line represents the median of the diurnal while the shaded region represents the interquartile ranges from 25-75th percentiles.**





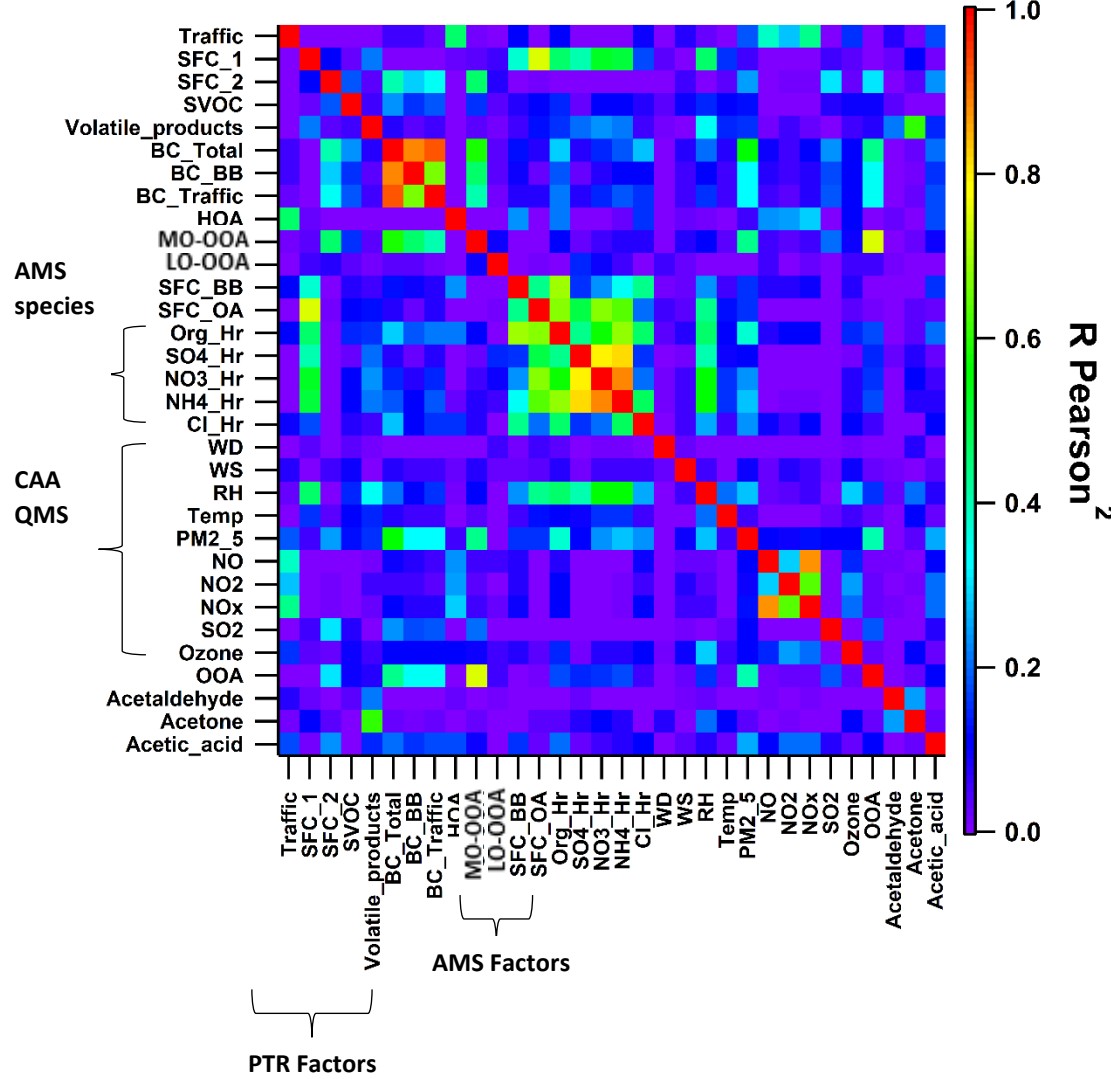

**Figure 6: Corelation of the five factors to the external measurements, including factors from AMS, Organics NR-PM₂.₅ and Inorganics NR-PM₂.₅, Black carbon (BC total, % BC from fossil and non-fossil fuels), CAAQMS data, total oxygenated organic aerosols (OAA), VOCs species. The CAAQMS data includes wind direction (WD), wind speed (WS), relative humidity (RH), ambient temperature (Temp), Particulate matter (PM2.5), nitric oxide (NO), nitrogen dioxide (NO₂), nitrogen oxides (NOₓ), sulphur dioxide (SO₂) and Ozone. The correlation between the timeseries of the parameters is represented by R Pearson², colour coded with rainbow color scheme, showing violet as 0 (no correlation) and red as 1 (highest correlation).**









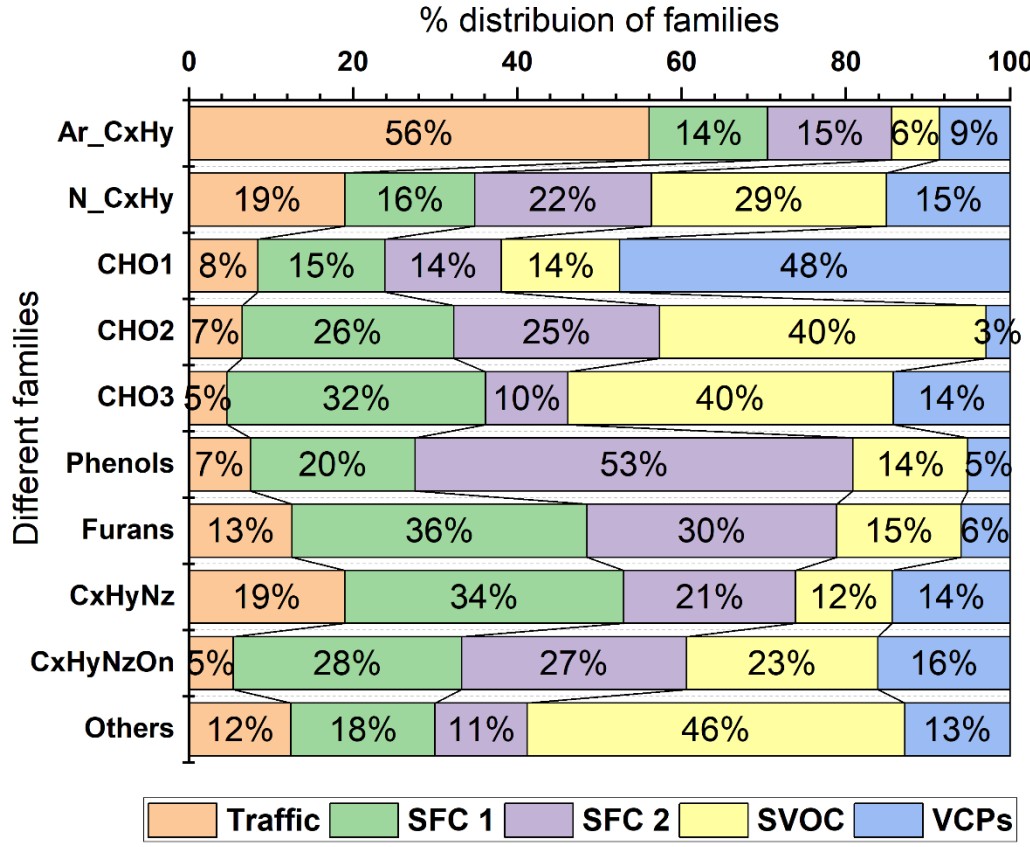

**Figure 7: Relative contributions (%) of different families to the individual factors**















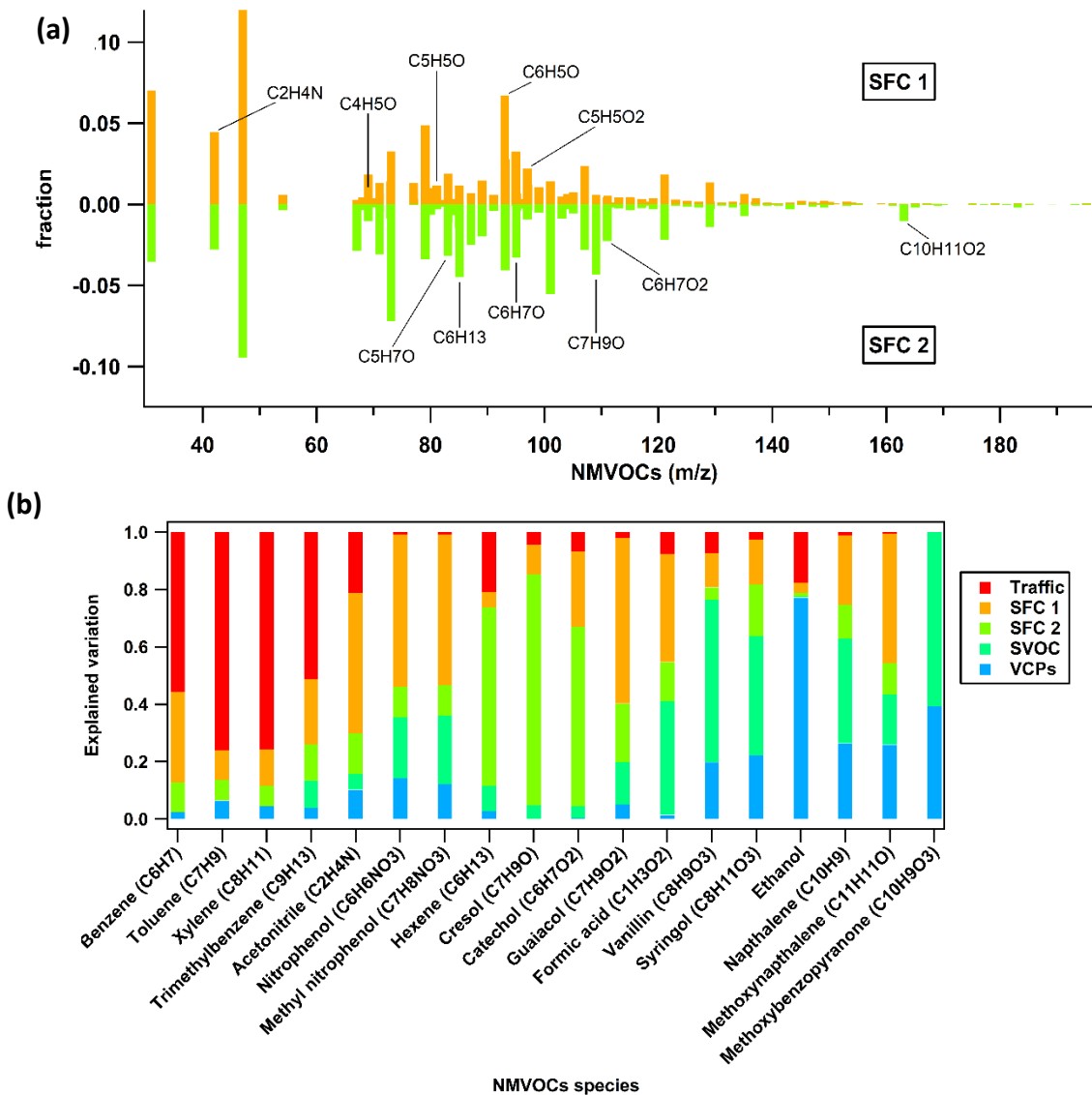

**Figure 8: (a) Comparison of relative composition of two factor profiles (SFC 1 and SFC 2). SFC 1 spectrum on top and SFC 2 spectrum on bottom. (b) Explained variation of selected NMVOCs species, stacked such that total explained variation is 1, colour coded by the five factors.**

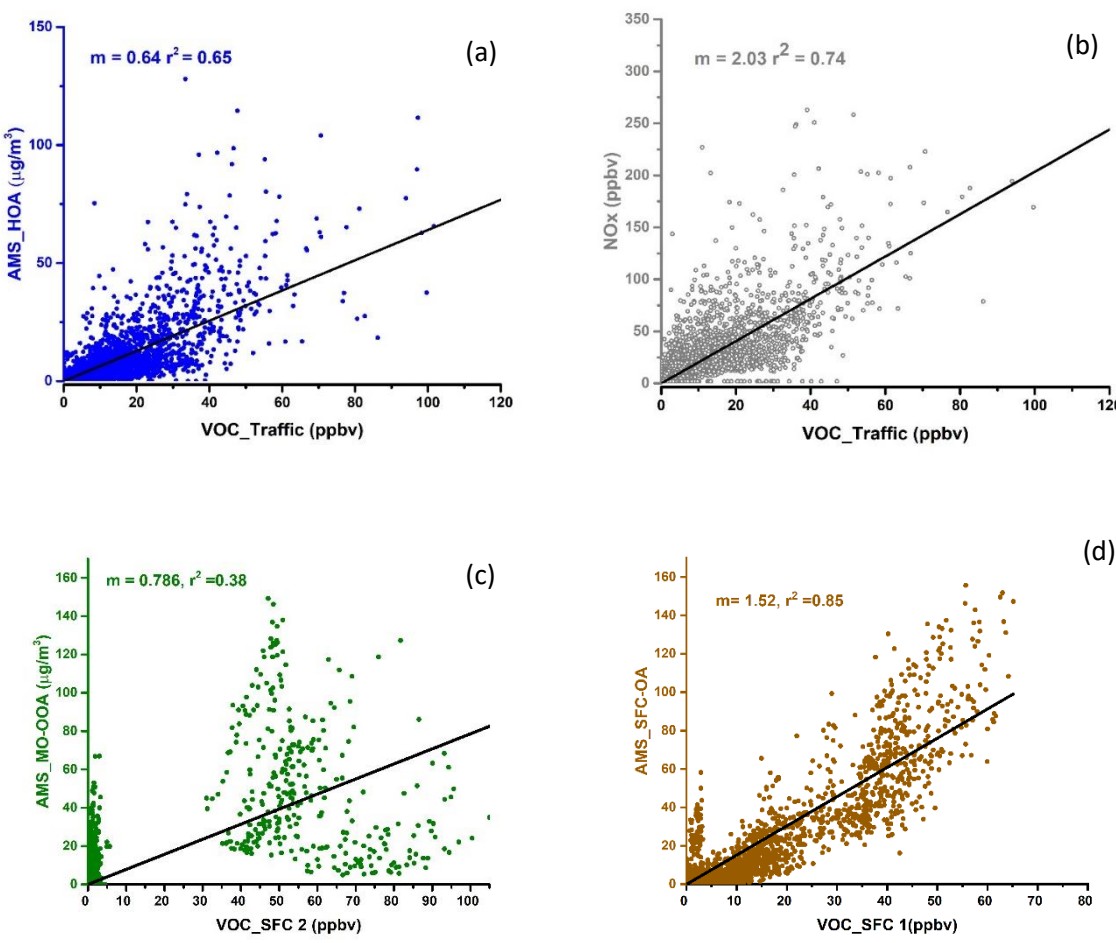

**Figure 9: Scatter plots showing a correlation between VOC_factors with their respective AMS_factors**

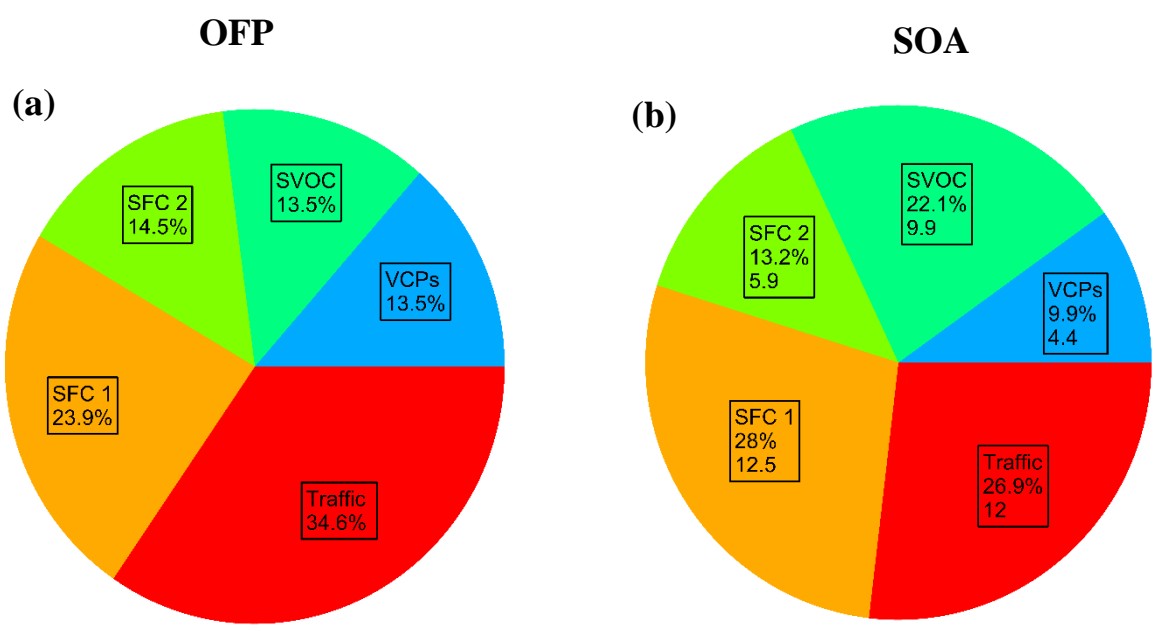

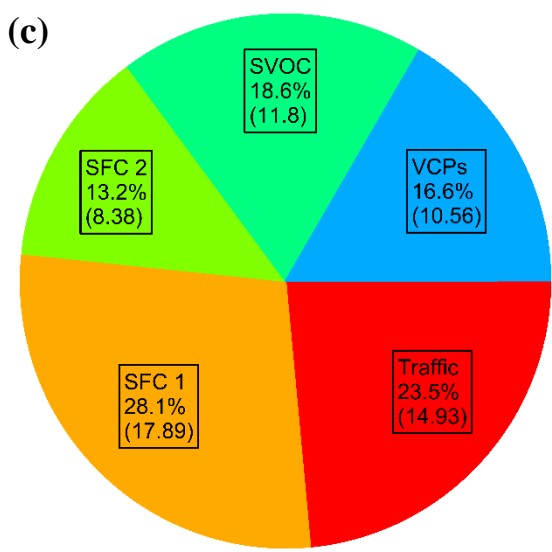

**Figure 10: Distribution in percentage (%) of individual factors to (a) Ozone formation potential (OFP), (b) SOA formation, (c) Relative contribution. The bottom absolute values (in brackets) for (b) and (c) are the SOA yield mass concentration (μg/m$^3$) and average mixing ratios (ppbv)**

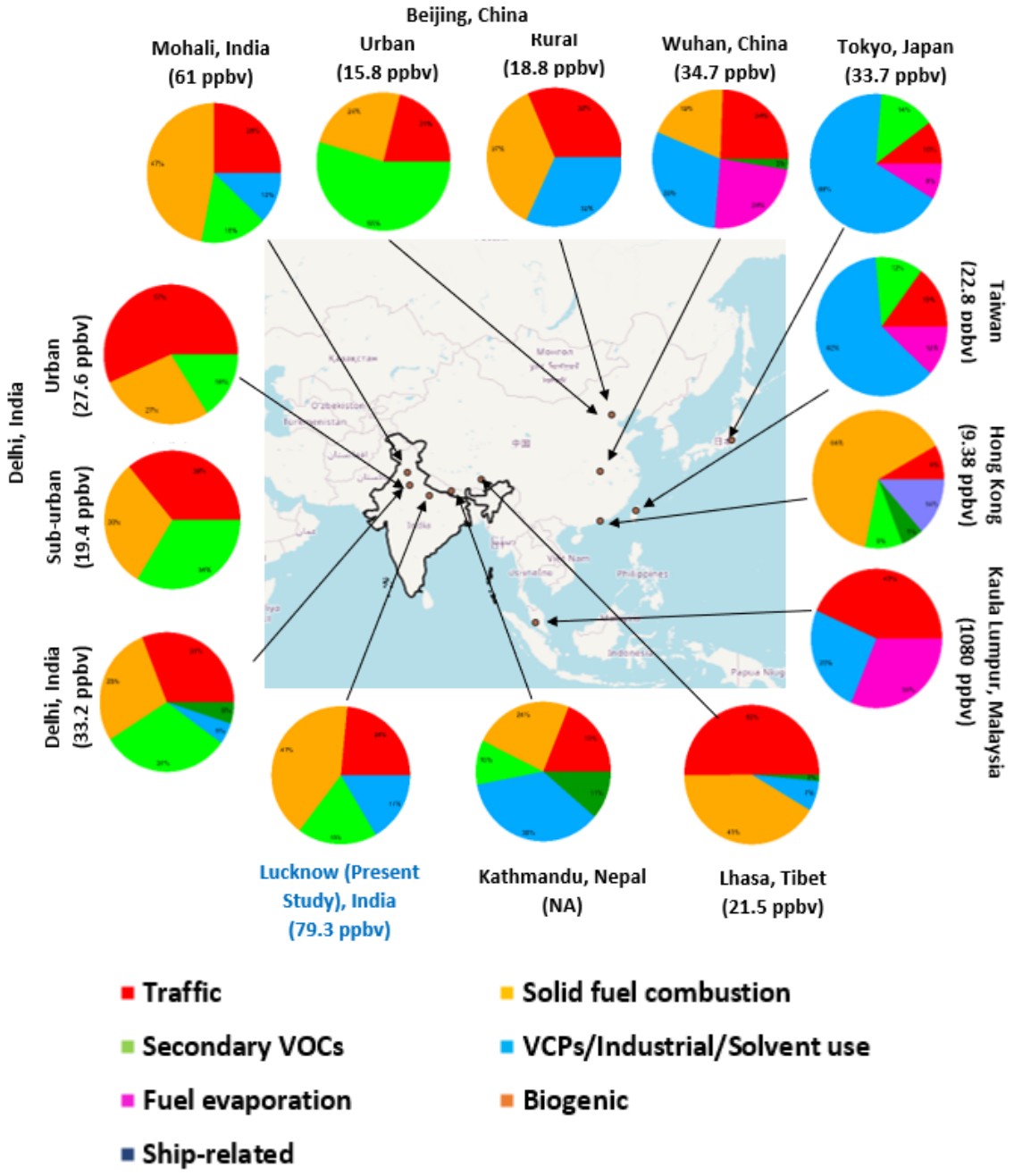

**Figure 11: Mapped- Pie charts showing various sources of NMVOCs in different Asian and Indian cities. The bottom values (in brackets) represent the averaged mixing ratios of total NMVOCs in respective study.**
