# Peer review of "Real-time measurements of NMVOCs in the central IGB, Lucknow, India: Source characterization and their role in O3 and SOA formation"

_EGUsphere, 2022_

## Author Response (AR1)

**Author Responses/Comments (text in blue) to comments by Referees (text in black) (AC 1 to RC 1)**

We thank the referees for their valuable comments which have greatly helped us to improve the manuscript. Please find below our point-by-point responses (in blue) after the referee comments (in black). The changes in the revised manuscript are written *in italic.*

**Comment on egusphere-2022-1165**
Anonymous Referee #1

General Comments

This study describes the deployment of a state-of-the-art Proton Transfer Reaction Mass Spectrometer instrument for high time resolution measurements of a large range of volatile organic compounds in an urban location in India. Positive Matrix Factorization was used to apportion the sources of the measured compounds. Relationships with simultaneous measurements of PM2.5 aerosol chemical composition via High Res ToF-AMSas well as Black Carbon (BC), NOx, SO2, Ozone, meteorological parameters and back trajectory analysis were used to support the selected PMF solutions and explore temporal variations.

The ozone formation potential and SOA yield of individual VOCs as well as each identified factor were estimated. This study identified traffic, solid fuel combustion, secondary VOC formation and volatile chemical products associated with industry as the dominant sources of VOCs at the sampling site. VOCs associated with traffic and solid fuel combustion had the highest ozone formation potential and estimated SOA yield.

This work is an addition to other recent studies using HR-PTR-MS in Ahmedabad (e.g. Sahu et al 2015, 2016, 2017) and Delhi (e.g. Wang et al 2020, Tripathi et al 2022, Jain et al 2022) with associated studies using a HR-ToF-AMS (Shukla et al 2021, Lalchandani et al 2021, Tobler et al 2020) undertaken with the researchers from the Indian Institute of Technology Kanpur. While the present study is of relevance to national and regional air quality management and population health studies, the manuscript requires further work to demonstrate novelty and impact for the wider atmospheric chemistry and physics domain. In particular:

- Co-located measurements by Hr-ToF-AMS and PTR-ToF-MS are uncommon and offer a novel opportunity to characterise total atmospheric organic carbon and relationships between gas and aerosol species. A more full presentation of the AMS measurementsand an exploration of relationships between gas and aerosol organics would significantly enhance the novelty of this manuscript.

- More in depth comparison with similar previous studies in Indian / other Asian cities would enhance the wider impact of this manuscript – synthesise common factors and their key trace species in VOC and NR-PM2.5 composition emerging from

these studies.What are the common factors identified in these studies, what unique factors emerge inindividual studies from different regions. The work by Zhang et al 2007 (https://doi.org/10.1029/2007GL029979) may provide a useful example of synthesis.

- While the purpose of these studies is to understand drivers of poor air quality and impact on population health, this is not highlighted in the discussion. The concentrations of individual air toxic VOCs (e.g. benzene, formaldehyde) could be compared to National / WHO air quality objectives. The relative contribution of SOA to total PM2.5 burden could also be presented. Identify days of poorest air quality – what factors were the major contributors?

- Some measurement issues need to be better addressed in the Methodology and Supplement to provide confidence in the interpretation (see below)

- Discussion on the influence of meteorological conditions and photochemistry associatedwith transport and aging on the airmasses sampled needs to be expanded.

- Manuscript requires a general proofread and some statements require supporting references. Be aware of use of past and present tense.

  Reply: We thank the referee for the constructive and critical comments. We thoroughly extended our sections on your suggestions with support from the literature. All your points have been addressed in below mentioned specific comments.

**Specific Comments**

Abstract

- Line 19 "the average concentrations of NMVOCs are relatively high during winter". Usequantitative statements ie NMVOCs were X-X% higher in winter than in summer
  Reply: We have added relative % in the statement as given below.

  *Line20-21*
  *The NMVOCs daily average concentrations were about ~30% high during winter months (December-February) than in summer (March-May).*

- Comment on which were dominant VOCs / VOC families that comprised NMVOC.
  Reply: We have also added information regarding the dominant VOC families in the text as follows:

  *Line 22-24*
  *The oxygenated volatile organic compounds and aromatics were the dominant VOC families throughout the period, accounting for ~57-80% to the total NMVOCs concentrations. Acetaldehyde, acetone and acetic acid were the major NMVOCs species present 5-15 times higher than other species.*

- Comment on % contributions of each factor to NMVOC.
  Reply: We have added percentage (%) in brackets to each corresponding source in

the abstract as given below.

*Line 26-28*
*They include traffic (23.5%), two solid fuel combustion factors: SFC 1 (28.1%) and SFC 2 (13.2%), secondary volatile organic compounds (18.6%) and volatile chemical products (16.6%).*

- Line 23 "Biomass burning contributes most of the NMVOCs and SOA formation, whileinterestingly traffic sources most influence ozone formation".
  Use consistent terms ie Biomass burning or SFC
  Reply: We thank the reviewer for the comment. We have changed the term to 'SFC' and checked the document for consistent terms also.

- Is the contribution of biomass burning factors to total OA (and SOA) from the AMS data reported?
  Reply: We represented and correlated the timeseries of AMS factors with PTR-MS factors in this manuscript. The detailed analysis related to organic aerosols from Hr-ToF-AMS are being prepared in a separate manuscript (Murari et al., in preparation) and (Patel et al., in preparation). Therefore, the contributions for different factors have not been reported from the AMS data.

- The potential SOA yield from the NMVOC factors was only estimated. Suggest rewording ie Aged and fresh emissions from Solid Fuel combustion (SFC 1 and 2) was the dominant contributor to total NMVOC and compounds related to these factors had a high SOA formation potential.
  Reply: We have edited the sentence as given below.

  *Line 28-30*
  *Aged and fresh emissions from Solid Fuel combustion (SFC 1 and 2) were the dominant contributors to total NMVOC and compounds related to these factors had a high secondary organic aerosols (SOA) formation potential.*

- Likewise for traffic sources and OFP.
  Reply: We have changed the line and added in the abstract as given below.

  *Line30-31*
  *Interestingly, traffic factor was the second highest contributor to total NMVOC and compounds related to this factor had high ozone formation potential.*

- Line 26 " The high temperature during summer leads to more volatilisation of oxygenated VOCs." Again be consistent – does oxygenated VOCs here refer to both SVOC and VCP factors or just VCP? Ie Higher temperatures in summer were associated with more volatilisation of oxygenated Volatile Chemical Products from industry sources.
  Reply: We thank the reviewer for this point. We rephrased the sentences in the abstract. We also checked the document for the consistent use of terms.
  Yes, it is observed that most of the NMVOCs associated with the factor VCPs are related to industrial sources. Moreover, it is also observed that the location of some of the solvent-based industries and textile industries are present in the close vicinity of the sampling site. Therefore, it is concluded that higher temperature during

summers were associated with more volatilization of these oxygenated VCPS from nearby industrial sources.

*Line 33-34:*
*The high temperature during summer leads to more volatilisation of oxygenated VOCs, related to VCPs factor.*

- The significance/ specialty of the study needs to be highlighted in the abstract – what isits practical significance (ie to the atmospheric measurement community), what is its significance to understanding /management of air quality in this region?
  Reply: We added the novelty and significance of the study in the abstract.

  *Line 34-37*
  *The study is the first attempt to highlight the sources of NMVOCs and their contribution to secondary pollutants (SOA and O3) formations in Lucknow city during winter and summer seasons. The insights from the study would help various stakeholders in managing primary and secondary pollutants within the city.*

**Introduction**

- Suggest presenting only a summary of direct and indirect effect of VOCs on air qualityas these will be familiar concepts to most ACP readers. Dedicate more of the introduction to summarising previous studies of NMVOCs and NR-PM2.5 composition inIndian and other Asian cities.
  Reply: We added the summary of the previous studies of NMVOCs in Indian context in the introduction. We also added one more section (Section 4) to the manuscript related to comparison with Indian and Asian cities, focusing on NMVOCs composition, their sources and relative composition.

[revised manuscript text omitted]

**Methodology**

Sampling Site Description

- Move details of inlet and instrument into next section.
  Reply: We moved the information to the next section.

  *Line 186-188*
  *During The study, the PTR-ToF-MS instrument's inlet is connected to a Teflon PFA (perfluoroalkoxy) tube (1.5m in length) for drawing air samples at the flow rate of 60mL/min.*

- Were met paramters measured at the site? Provide more detail on meteorology differences between seasons temp, RH, for this location.
  Reply: The sampling site (building) is a part of the government CAAQMS (central Ambient air Quality Monitoring station), maintained by CPCB (Central Pollution Control Board). Therefore, these met parameters are not directly measured but the data have been downloaded from the government portal (https://app.cpcbccr.com/ccr/#/caaqm-dashboard-all/caaqm-landing) for the station Gomti Nagar, Lucknow. We believe as the sampling site was maintained in the same building where CAAQMS station is located, those parameters can be used in our analysis. Along with it, we have also added more information about the meteorological parameters across seasons.

  *Line 155-164*
  *The study period is divided into two seasons according to the classification by IMD (Indian Meteorological Department) as winter (Dec-Feb) and summer (March-May). The gaps in the sampling period from 3-8th January and 21st March - 9th April were due to maintenance and calibration of the instrument. The average daily temperature is ~28 ºC over the whole study period in the city. The mean daily temperature during winters (Dec-Feb) was around ~25±2.5 ºC and during summers (March-May) around 32±3 ºC. The relative humidity ranged from 64±14% during winters and 42±11% during summers. The comparison of temperature and relative humidity changes during both seasons are shown as box plots in supplementary Figure S1. These values are based on the days when NMVOCs measurements exist. The pre-dominant wind direction was South-Southeast during colder and Southwest during warmer periods, as shown in supplementary Figure S1.*
  *Supplementary Figures S1*

[Figure]

**Figure S1: (a) Wind rose plots showing wind speed (m/sec) and wind direction in different seasons, winter and summer at the sampling site. (b) Box plots showing the temperature variation during the different seasons, summer and winter (c) Similarly, box plots showing the variation of the relative humidity during the different seasons summer and winter. The whiskers showing the 25-75th percentile of the data, while red dot represents the average temperature of the season.**

**Instrumentation and data analysis**

- Suggest rename to PTR-ToF-MS measurements of NMVOCs
  Reply: We have changed the section's name.

  *2.2. PTR-ToF-MS measurements of NMVOCs*

- Suggest further reducing general info on PTR-ToF-MS method and focus on presenting specific measurement details.
  - More inlet detail – what was the inner diameter of the PFA inlet ? Was the flow down this inlet 60ml/min total or did the PTRMS just sub-sample 60ml/min from a higher sample inlet flow? ie what was the residence time in inlet?
    Reply: The inner diameter of the inlet was 0.75 mm. There were many instruments

connected to the main inlet and PTRMS just sub-sample 60ml/min from a higher sample inlet flow. The residence time of the air in the inlet was less than 1 sec.

*Line 186-189*
*During the study, PTR-ToF-MS instrument's inlet was connected to a Teflon PFA (perfluoroalkoxy) tube (1.5m in length) for drawing air samples at the flow rate of 60mL/min. The inner diameter of the tube was 0.075mm and the residence time of the air in the inlet was less than 1 sec.*

- Provide specific info on calibration and zero measurements here and in supplementie mean +- stdev of compound specific sensitivities, how were background corrections applied, range of MDLs ?
  Reply: The calibration set up and details including zero measurement are already discussed in our previous study. We have included more information in the supplement according to your suggestion. For MLD, we have calculated the MLD using 3σ (standard deviation) of the zero air of 20 min time duration data points.

**Calibration and background**

A certified standard gas mixture (L5388, Ionicon Analytik GmbH Innsbruck, with a stated accuracy better than 8%) containing ~1.0 ppmv of VOCs (methanol, acetonitrile, acetaldehyde, ethanol, acrolein, acetone, isoprene, crotonaldehyde, 2-butanone, benzene, toluene, o-xylene, chlorobenzene, α-pinene, and 1, 2-dichlorobenzene) was used for the calibration. Calibrations of mentioned VOCs were performed using dynamic dilutions of the standard gas mixture. Fig (xx) depicts the setup of a gas calibration unit (GCU) (GCU-advanced v2.0, Ionicon Analytik GmbH, Innsbruck, Austria)) that was used for the multipoint calibration and determination of the blank levels of different VOCs. GCU consists of two mass flow (MFC) controllers to control the flow rate of standard gas (MFC-std) and dilution gas (zero-air, MFC-dil). For the calibration of VOCs (dynamic dilution) and determination of the blank level (background), the flow rate of zero-air was set at ~500 sccm/min. The background measurements were performed using a dry zero-air cylinder every 2 weeks. The background (blank) signals were already taken into account to calculate the mixing ratios of VOCs during the data extraction. Further details are discussed in our previous studies (Sahu and Saxena, 2015; Sahu, Yadav and Pal, 2016a).

*Line 202-205*
*A detailed description of the calibration set-up and details including zero measurement is given in the previous studies (Jain et al., 2022; Tripathi et al., 2022), given in supplementary Figure S2. For method detection limits (MDL), we have calculated the MDL using 3σ (standard deviation) of the zero air of 20 min time duration data points.*

[Figure]

- Did you use measured sensitivity or calculated sensitivity based on k rates and transmission for all VOCs reported or only those not in cal std? comment on uncertainty associated with these approaches?
  Reply: We have calculated sensitivity for standard using calibration and also calculated sensitivity based on k rates and transmission for VOCs those are not in cal std. The total uncertainty is in the range of 8%–13% in the calculations of the mixing ratios of VOCs include the uncertainties in the mass flow controllers (MFCs) of GCU and standard mixture (±5%–6%). It is also reported in our previous study Sahu & Saxena, 2015; Tripathi & Sahu, 2020.

- Why were formaldehyde and methanol excluded from compounds reported? These were likely to be very significant.?

  Reply: We agree that these NMVOCs are significant in the ambient atmosphere. But for the PMF analysis, these compounds are excluded. These ions are 5-15 times higher than the other ions due to their natural abundance (background concentrations) and high emission rates. However, these ions are minor contributors to SOA formation and only substantially contribute to the formation of ozone, which is a major issue in summer. It is also observed in our previous studies (Jain et al., 2022) that including these ions in PMF analysis, will produce solutions/factors where only these ions are well-explained.

- PTRMS sensitivity to formaldehyde is low (quasi-thermoneutral) and humidity dependent – applying a standard k-rate approach to this species is erroneous – use empirically derived, humidity dependent sensitivity factors or exclude from your analysis.
  Reply: Thank you for your suggestion. I agree with reviewer and we have excluded

formaldehyde from our analysis.

- ▪ Acetic Acid is notoriously difficult to measure due to its stickiness – how did you account for this? How were the data analysed (PTRMS viewer), were the data reported here averaged?
  Reply: Yes, we agree with reviewer that it is notoriously difficult to measure due to its stickiness. Here, the inlet line is heated at 60 degrees ºC temperature. The previous studies were reported the acetic acid at the same environmental conditions for Delhi region using PTR-ToF-MS (Wang et al. 2019, Tripathi et al. 2022). We have analyzed the data using PTRMS viewer 3.4, with a time resolution of 30 seconds. Yes, the data reported in this study were averaged to 15 minutes for PMF analysis.

**Supporting measurements**

- ▪ Suggest separate section (2.3) HR-ToF-AMS measurements of aerosol composition providing more detail on HR-ToF-AMS measurements – aerosol inlet, calibrations, acquisition parameters, PMF followed by 2.4 Supporting measurements – BC, NOx, O3, SO2, meteorology
  Reply: We thank the reviewer for the suggestions. We have added the details about the instrument, its setup and data analysis including all of the points (aerosol inlet, calibrations, acquisition parameters, PMF) mentioned in the section 2.2, related to HR-ToF-AMS. The detailed analysis and results of PMF are being prepared in another manuscript by our group (Murari et al., in preparation). Therefore, they are beyond the scope of this study. We added the section 2.2 and 2.3 dedicated to HR-ToF-AMS measurements and supporting measurements, respectively. Please find the details below.

  *Line 209-257*

[revised manuscript text omitted]

- The authors provide a good explanation of the PMF approach used and logic forselecting optimum solution.
  Reply: We thank the referee for the constructive comment and appreciation.

- Significant text is dedicated in this section and section 3.2.1 to describing the PMF method and optimum solution selection process. Suggest merging these sections intoone (sect 2.4) and supplement if text limited. Focus results and discussion on the characteristics and behaviour of the selected factors.
  Reply: We have merged the two sections and also have moved the part of the section 3.2.1 into supplementary. Please see below for further details.

*Line 296-324*

[revised manuscript text omitted]

**Ozone formation potential and SOA yield of NMVOCs**

▪ Details on the processes driving O3 formation and their non-linearity were discussed in the Intro and do not need to be repeated here. Only details relevant to the OFP estimation method should be presented. The paper that first described this method should be referenced in the opening sentence.

Reply: We have removed the sentences related to the processes driving O3 formation from the section. We have also added the reference of the method as given below.

*Line 327-328*
*Ozone formation potential (OFP) is a reactivity-based estimation technique to assess the sensitivity of the VOCs for ozone formation (Carter, 1994, 2010).*

▪ Describe MIR. How many of the species reported have an MIR – does excluding those without MIR bias the results?

Reply. The maximum amount of ozone produced after small increment of VOCs into a representative atmospheric system is known as maximum incremental reactivity (Carter 2010). We agree that there are only 40 compounds for which MIR values are available and included in the analysis. The rest compounds have been observed to contribute to ozone formation not at significant levels. Therefore, excluding those compounds without MIR would not bias the results.

*Line 331-334*
*This approach is based on calculating OFP using maximum incremental reactivity*

*(MIR) values for individual VOC species, reported and updated by (Carter 2010).. MIR values are calculated as the change in ozone formed by adding a VOC to the base case in a scenario with adjusted NOx concentrations. OFP of individual VOCs are estimated using Equation 4.*

$$OFP(j) = [VOC_j] \times C_j \times MIR_j \qquad (4)$$

- Line 236 "the literature determines the SOA yield" – revise sentence. The SOA yields reported by Bruns et al 2016 were used for this analysis. Where SOA yields were not knoe, compounds with >6 carbons were assumed to have an SOA yield of 0.32 basedon ….?

  Reply: we have changed the sentence. The average value of SOA yield of about 18 similar compounds (C >6) is used as an individual value for unknown compounds. The compounds having more than 6 carbon atoms are considered to contribute to SOA significantly (Bruns et al., 2016). The text in the section is changed as given below.

  *Line 356-361*

  *The SOA yields $Y_{SOA(j)}$ reported by Bruns et al 2016 were used for this analysis. The compounds for which SOA yield values are not available from the literature directly, it is estimated that compounds having carbon atoms more than 6 (C >6) will have the same SOA yield of 0.32 (Bruns et al., 2016). Based on their structure, the compounds (C >6) are considered to contribute significantly to SOA (Bruns et al., 2016). The average value (0.32) of the published SOA yield of 18 compounds (C>6) is used. In this study, the individual SOA yield values considered are given in the supplementary Table S3.*

- Note this analysis represents the OFP and SOA formation potential of the air mass composition at the sampling site and not the OFP SOA FP of the various emission sources. Ie airmasses dominated by fresh emissions (eg traffic) will have a differentOFP and SOA FP than aged airmasses (eg long range transport of BB plumes)

  Reply: We have added the following lines in the section text.

  *Line 368-371*

  *This analysis represents the estimated OFP and SOA formation potential of the air mass composition at the sampling site, not the actual OFP and SOA formation potential from various sources. This means that airmasses dominated by fresh emissions (e.g., traffic) will have a different OFP and SOA formation potentials than those in aged airmasses (e.g., long range transport of BB plumes) or any other source.*

**CWT back trajectory analysis**

1) Rename section "Concentration weighted back trajectory analysis"
Reply: We renamed the mentioned section.

*Line*
*Section 2.7. Concentration weighted back trajectory analysis*

2) 100m of arrival height is repeated x2 in the text.
Reply: We removed the repeated text.

3) Acknowledge who this method was first described by.

Reply: We have added the references in the section.

*Concentrated-weighted backward trajectory (CWT) analysis determines the originating source and transport of air parcels at the receptor location within a specific period (Seiber et al., 1994, Draxler et al., 1998).*

4) Note the reader may require explanation to reconcile why prevailing winds during study as shown in windroses in Fig S1 were predominantly from the SE – SW yet the CWT plots in fig S3 show higher trajectory density from the North.

Reply: We agree with the reviewer that there is a difference between the dominant wind direction when using meteorological data from CAAQMS at Gomti Nagar, Lucknow (Source of data: CPCB portal, https://app.cpcbccr.com/ccr/#/caaqm-dashboard-all/caaqm-landing) and plotting CWT trajectories using the HYSPLIT model and GDAS files (Source of data: NOAA, ftp:/arlf tp.arlhq.noaa.gov/pub/archives/gdas1). We believe that these differences could be due to the difference of measurement method in collecting these data. The wind direction data from CPCB is measured using the automatic weather station which is installed on the UPPCB building (~10m from the ground). This is possible that this data is more dominant by the local wind conditions. The CWT trajectories are plotted using met data collected 100 m above the ground. This represents the overall regional wind direction. Although, for the source of VCPs factor, the dominant direction is South, from both CAAQMS data and CWT plots (Supplementary Figure S3).

**Results and Discussion**

3.1 NMVOC concentrations and temporal variation.

1) In general, this section requires significant revision to clarify the aim of this section and the concepts presented to improve interpretation. Suggest here presenting

- summary stats for NMVOC, dominant species (Acetald., Acetone, Acetic Acid) and relative contributions of these species and each of the VOC families to NMVOC.

- seasonally and diurnally varying patterns.

Reply: For the above comment, We revised the whole section and added more discussion. We also compared the seasonal and diurnal pattern of total NMVOCs, and three dominant species as given below.

*The average daily concentrations of measured NMVOCs during the study period was 125.5 ± 37.5 ppbv. Figure 2 shows the daily time series and monthly mean concentrations of NMVOCs, inorganics and organics fractions of Nr-PM2.5, O3, NOx, SO2, temperature, relative humidity, wind speed, and wind direction. Out of 173 detected NMVOCs, the level of three major species (Acetaldehyde, Acetone, and Acetic acid) were present 5-15 times higher than for other species, as shown in supplementary Figure S2. The monthly averaged concentrations of NMVOCs were observed relatively higher during winter months from December (193.7 ppbv) to January (110.2 ppbv) till February (109.7 ppbv) than- during the summer months, March (101.2 ppbv), April (137.8 ppbv) and May (150.8 ppbv). The averaged concentrations of NMVOCs (127±40 ppbv), as well as Nr-PM2.5 (inorganics and organics) (102.8±51 μg/m³) was higher, during the winter months.. The calm conditions and relatively*

*lower planetary boundary layer during winters have slowed down the dispersion of the pollutants. In contrast, during the summer months, Nr-PM2.5 (39.8 ±20 µg/m³) decreased drastically, but NMVOC concentrations (122 ±32 ppbv) were similar to winters. This may be due to high temperatures during warmer periods may lead to more photooxidation of primary VOCs (Sahu, Tripathi and Yadav, 2017), production of biogenic VOCs (Baudic et al., 2016; Sahu, Tripathi and Yadav, 2017) and evaporation of volatile household products (Qin et al., 2021). While aerosol particles managed to disperse in the atmosphere due to high planetary boundary layer and windy conditions. The difference of characteristics of the emission sources during both seasons may have also played an important role.*

*The three most abundant NMVOCs were not considered in the PMF analysis as explained in section 2.5. The remaining 170 NMVOCs considered for the PMF analysis varied from m/z 42. 034 to m/z 197.216 The average concentrations of these 170 NMVOCs was 79.3 ±30.6 ppbv. The averaged concentrations during winters were 86.7±35 ppbv, a bit more than during summers as 68.3±17.2 ppbv. These NMVOCs belong to different families based on their chemical composition. They are categorised as aromatics (Ar_CxHy), simple non-aromatics (N_CxHy), furans (Furans), phenols (Phenols), oxygenates: first (CHO1), second (CHO2), and third order (CHO3), nitrogen-containing compounds (CxHyNz and CxHyNzOn) and others. The others include high-order oxygenates (CHO4) and some hydrocarbons (CxHy). The degree of unsaturation (i.e. the number of rings and/or double bonds) of more than 4 distinguishes aromatics (ArCxHy) from the CxHy family. This allowed us to identify important VOCs markers, their families, and their role in their atmospheric chemistry. Overall, during the study period, highest contributing family belongs to oxygenates and aromaitcs. The CHO1, CHO2 and CHO3 families were 28.8% (~20.1 ppbv), 16.8% (11.7 ppbv), and 2% (1.4 ppbv) of total NMVOCs concentrations. The contribution from Ar_CxHy, and N_CxHy were about 21.5% (~15 ppbv), and 10.6% (~7.4 ppbv), respectively. Nitrogen containing compounds were relatively less present (5.6 % CxHyNz and 1.2% CxHyNzOn). 6.3% (~4.4 ppbv), and 3.7% (~2.6 ppbv) were contributed by Furans and Phenols at the site, the rest was included in others (3.4%). The CPCB notified the annual National Ambient Air quality Standards (NAAQS) only for benzene as 5 µg/m3 (~1.6 ppbv). While WHO recommended no safe level of exposure of benzene. The mean mixing ratio of benzene during the study period found to be 2.9 ±1.9 ppbv which is around 2 times higher than the standard guidelines. Prolonged exposure or high short-term exposure to benzene adversely affect the health of citizens of the city due to its haematotoxic, genotoxic and carcinogenic properties.*

2) Figure 2 – include lines to indicate winter and summer periods. Note low data capturefor months to Dec and to April may bias these results. In top panel 'VOC time series' Consider instead of plotting 'Other" plot rel. contribution of VOC families

Reply: We added time series of families to the Figure 2 as shown here. We also added the winter and summer periods by indicating the line diving into the two seasons. We took the average on daily basis, which does not bias the results.

[Figure]

**Figure 3: Daily averaged time series of acetaldehyde, acetone, and acetic acid, other NMVOCs, PM$_{2.5}$ and its organic fraction, NO$_2$, SO$_2$, O$_3$, temperature, relative humidity, and wind speed and direction**

3) Line 264 – 266 "the highest concentrations of NMVOCs, NR-PM2.5 during the winter months infer their common sources" – meteorology would also play an important role ie calm conditions in winter and lower PBL? Use quantitative statements ie provide NMVOC and NR-PM2.5 concs in brackets.

Reply: We have added the averaged concentrations of NMVOCs and Nr-PM$_{2.5}$ based on the seasons and have revised the sentences as follows.

*Line 388-399*
*The monthly averaged concentrations of NMVOCs were higher during winter months from December (193.7 ppbv) to January (110.2 ppbv) till February (109.7 ppbv) than during the summer months, March (101.2 ppbv), April (137.8 ppbv) and May (150.8*

*ppbv). The averaged concentrations of NMVOCs (127±40 ppbv), as well as Nr-PM2.5 (inorganics and organics) (102.8±51 µg/m3) was higher, during the winter months.. The calm conditions and relatively lower plentary boundary layer during winters have slowed down the dispersion of the pollutants. In contrast, during the summer months, Nr-PM2.5 (39.8 ±20 µg/m3) decreased drastically, but NMVOC concentrations (122 ±32 ppbv) were similar to winters. This may be due to high temperatures during warmer periods may lead to more photooxidation of primary VOCs* (Sahu, Tripathi and Yadav, 2017*), production of biogenic VOCs* (Baudic *et al.*, 2016; Sahu, Tripathi and Yadav, 2017*) and evaporation of volatile household products* (M. Qin *et al.*, 2021)*. While aerosol particles managed to disperse in the atmosphere due to high planetary boundary layer and windy conditions. The difference of characteristics of the emission sources during both seasons may have also played an important role.*

4) Line 266 – "In contrast, during the summer months, PM2.5 decreases drastically, but NMVOC concentrations are relatively highest, implying additional sources of NMVOCs". –sentence requires revision.

Reply: We have revised the sentence. Please look at the previous reply.

5) Line 275 "diurnal variations of secondary formation, anthropogenic emission level, weather and PBL heights can be explained by OVOC/ benzene ratios to some extent" revise sentence – the ratio of OVOCs/benzene does not explain these factors.
Reply: The sentence has been revised as given below.
*Line 425-427*
*Diurnal variations of secondary formation, anthropogenic emission level, meteorological conditions, and PBL heights influence OVOCs/benzene ratios* (Sahu, Yadav and Pal, 2016b; Sahu, Tripathi and Yadav, 2017; Tripathi *et al.*, 2022) *to some extent.*

**3.2 PMF results**

3.2.1 Optimum solution selection

Merge this with discussion of PMF methodology in 2.4.
Reply: We have merged the two sections and also have moved the part of the section 3.2.1 into supplementary.

3.2.2 Profile and diurnal variation

1) Suggest rename / restructure this section '3.2 Characteristics of selected PMF factors'
Reply: We have renamed the section.

4) General info on outcomes of PMF, outline info that will be presented to characterize each factor and then present detail under Sub-headings 3.2.X for each factor'

Reply: We added the sub-heading for each factor as suggested by the reviewer. We also added discussion for the respective factors, markers, and other details in each sub-section.

*Line 463-611*
*3.2.Characteristics of selected PMF factors*
*This section includes a discussion of the selection of the source apportionment solution and its*

*interpretation. The NMVOCs factors are identified based on their mass spectra, diurnal and temporal variation, and correlation with external tracers. For the first time, we have included mass spectra of 170 NMVOCs from m/z 42.034 to m/z 197.216 in the PMF analysis. The three abundant NMVOCs (compounds below m/z 42) detected by PTR-ToF-MS, acetaldehyde, acetone, and acetic acid, are not included in PMF analysis. Including these NMVOCs in the PMF analysis resulted in biased solutions where only these ions are well-explained. Additionally, a few small alkanes and alkenes (C1-C4) compounds, which are not detected by PTR-ToF-MS, are excluded from PMF analysis. However, previous studies have found that these ions are minor contributors to SOA formation. Included compounds (above m/z 42) are major contributors to SOA formation and dominant markers of various sources. As explained in section 2.4, the optimum solution after the PMF analysis chosen is a 5-factor solution. This selected 5-factor PMF solution exhibits distinct mass spectral characteristics related to different sources and atmospheric processes. Figure 5 shows the intricate plots of the profile and diurnal variation of the 5-factor solution. The five factors are Traffic, SFC 1 (solid fuel combustion), SVOC (secondary volatile organic compounds), SFC 2, and VCPs (volatile chemical products) after thoroughly investigating markers, chemical species and their families, diurnal variation, and relation to meteorological parameters and external measurements. The diurnal variation of the factors for two seasons (winter and summer) were compared, as shown in supplementary Figure S7. The timeseries of the five factors resolved from NMVOCs mass spectra are co-related with external measurements such as oxygenated organic aerosols (OOA), Black carbon (BC) concentrations, CAAQMS data (WD, WS, RH, Temp, NO, NO2, NOX, SO2, O3) as given in Figure 6.*

**3.2.1.  *Factor 1: Traffic**

**3.2.2.  *Factor 2: Solid Fuel Combustion (SFC 1)**

**3.2.3.  *Factor 3: Solid Fuel Combustion (SFC 2)**

**3.2.4.  *Factor 4: Secondary volatile organic compounds (SVOC)**

**3.2.5.  *Volatile chemical products (VCPs)**

2) Fig 5 – add formula/name to key peaks. Figures 6 and 7 are useful as is. Fig 8 – check AMS species labels are the same as presented in text (ie MO-OOA and LO-OOA). Figure 9c while r2 value is 0.52 the plot indicates a poor relationship.

Reply: We modified Figure 5 by adding the formulas to respective peaks and markers for each factor. In Figure 8, AMS species labels are changed, similar to text as MO-OOA and LO-OOA. There is a typing mistake in Figure 9, we corrected the r2 value as 0.38, which indicates their poor relation.\

*Revised Figure 5:*

[Figure]

**Figure 5: Profile and diurnal variation of individual factors of selected 5-factor solution after PMF analysis**

*Revised Figure 8:*

[Figure]

**Figure 8: Corelation of the five factors to the external measurements, including factors from AMS, Organics NR-PM$_{2.5}$ and Inorganics NR-PM$_{2.5}$, Black carbon (BC total, % BC from fossil and non-fossil fuels), CAAQMS data, total oxygenated organic aerosols (OAA), VOCs species. The CAAQMS data includes wind direction (WD), wind speed (WS), relative humidity (RH), ambient temperature (Temp), Particulate matter (PM2.5), nitric oxide (NO), nitrogen dioxide (NO$_2$), nitrogen oxides (NO$_X$), sulphur dioxide (SO$_2$) and Ozone. The correlation between the timeseries of the parameters is represented by R Pearson$^2$, colour coded with rainbow color scheme, showing violet as 0 (no correlation) and red as 1 (highest correlation).**

[Figure]

**Figure 9: Scatter plots showing a correlation between VOC_factors with their respective AMS_factors**

3) Add time series of factors – consider adding to Fig 2 to aid comparison with other variables.

Reply: We added time series of factors to the Figure 2 as shown here.

[Figure]

**Figure 3: Daily averaged time series of acetaldehyde, acetone, and acetic acid, other NMVOCs, PM$_{2.5}$ and its organic fraction, NO$_2$, SO$_2$, O$_3$, temperature, relative humidity, and wind speed and direction**

5) Use consistent presentation of characteristics for each factor

- Factor identification ie traffic, SFC1, SFC 2

- Marker species and their average % contribution to the factor.

- relationships to other atmospheric species – move discussion from 3.3 under each

relevant sub-heading.

- diurnal / seasonal patterns which help identify sources eg diurnal patterns that align withpeak traffic; seasonal patterns of SFC.

- CWT plots for each factor – do they align with location of known / likely sources?

- comparison with previous studies – similar markers and % contributions?

Reply: We modified the sections as suggested by the reviewer. Each sub-section is added for each factor, discussing every point as follows, (1) Important markers, identified, their average % contribution. (2) Similar markers in previous studies (3) identified dominant families in each factor (4) diurnal and seasonal pattern of the factor (5) their correlation to other related external measurements and parameters (6) CWT plots, their relevance and possible source locations.

Line 484-

**3.2.1. Factor 1: Traffic**

*The first factor is identified as traffic. It is characterized by the presence of aromatics, such as benzene (m/z 79.053, $C_6H_6H+$), toluene (m/z 93.07, C7H8H+), xylene (107.09, C8H10H+), C9-aromatics (121.1, C9H10H+), and C10-aromatics (135.12, C10H14H+).56% of the total aromatics are explained by this factor, as shown in Figure 6. The explained variation of individual NMVOCs, such as C6H6H+, C7H8H+, and C8H10H+ by the traffic factor is around 0.56, 0.77, and 0.76, respectively, as shown in Figure 7 (a). The NMVOC's traffic factor shows a temporal correlation (Pearson r2 ~ 0.74) with nitrogen oxides (NOx), which is also an indicator of vehicular emissions (Figure 8). Also, this factor has a good correlation (Pearson r2 ~ 0.65) with the AMS_HOA (PMF-resolved factor from HR-ToF-AMS), as shown in Figure 9 (a). This AMS_HOA factor is characterized by aliphatic hydrocarbons, typically associated with traffic exhaust emissions (Lalchandani et al., 2021). It infers that vehicular exhaust is one of the common sources influencing the release of NMVOCs, NOx and primary OA. These NMVOCs and primary OA also exhibit similar diurnal variation, having sharp peaks during morning and evening hours, as shown in supplementary Figure S7.This diurnal pattern indicates the vehicular commute pattern in the city, high density of vehicles on the roads during rush hours in the morning and evening. The traffic factor in previous studies observed similar markers and diurnal pattern in Delhi (Jain et al., 2022; Wang et al., 2020) and Beijing* (Wang *et al.*, 2021b)*, indicating the similar commute pattern in most of the urban cities. Other source-specific studies also identified similar markers for vehicular emissions* (Caplain *et al.*, 2006; Cao *et al.*, 2016). *The back trajectory analysis of the factor, (CWT graph), given in Supplementary Figure S3, shows the probable sources of traffic present near the sampling site.*

**3.2.2. Factor 2: Solid Fuel Combustion (SFC 1)**

[revised manuscript text omitted]

3.4 OFP and SOA yield from individual sources

1) Is there a relationship between factors/ species with high SOA and O3 potential and measured concentrations of SOA and O3? Consider a time lag in peaks.

Reply: We thank the reviewer for the comment. When comparing the measured SOA, which is considered as the sum of more-oxidized oxygenated OA factor (AMS_MO-OOA) and one low-oxidized oxygenated OA (AMS_LO-OOA) factor, based on the previous studies with the five highest contributor NMVOC species such as benzene, toluene, xylene, phenol, and naphthalene. The high and low peaks are co-occurring during winters. The plots of time series have been added in the supplementary figure and text in the main section as given below. Similar analysis could not be reported for $O_3$ as the $O_3$ time series does not have any relation with NMVOCs concentrations, even after considering the time lags in the peaks. This may be due to ozone is only formed after two hours and would be found downwind of the site. Another issue is that local NO emissions decrease O3.

*Line 626-632*

*The measured SOA from HR-ToF-AMS may be considered as the sum of more-oxidised oxygenated OA factor (AMS_MO-OOA) and one low-oxidised oxygenated OA (AMS_LO-OOA) factor (Lalchandani et al., 2021). The five highest contributor to SOA formation potential were correlated with the measured SOA, in the supplementary figure S10. The high-resolution time series shows the co-occurrence of high and low peaks of benzene, toluene, and xylene with measured SOA during the day and night hours. This shows the significant role of aromatic NMVOC species in the formation of SOA.*

*Supplementary:*

[Figure]

**Figure S8: High-resolution time-series of measured SOA from HRr-ToF-MS and NMVOCs species measured from PTR-ToF-MS, which potentially contributed maximum to the formation of SOA.**

2) This section would be improved by comparison with previous studies and discussion on relevance of this section ie for control strategies to reduce O3 and SOA.
Reply: We thank the reviewer for the suggestions. We have improved the section by comparing the control strategies of ozone and SOA from previous studies. We have

included some previous studies for Asian cities. In the previous study, it has also been noticed that the contribution of aromatics (xylene and toluene) have substantial affect to the ozone formation potential for different cities of Asia including Delhi, Guangzhou, Beijing (Zheng et al. 2009, Duan et al. 2008, Tripathi et al. 2022).

*Line 618-621*

*Toluene, xylene, and isoprene were found to be the highest contributor in terms of OFP in other Asian cities, including Guangzhou and Beijing (Zheng et al., 2009; Zhu et al., 2016; Zhan et al., 2021; Duan et al., 2008). In the previous study in Delhi, it has also been noticed that the contributions of aromatics (xylene and toluene) have a substantial effect on the ozone formation potential (Tripathi et al., 2022).*

*Line 634-638*

*Previous studies have also found that aromatic hydrocarbons contributed more than 95% to the SOA formation potential in other Asian cities (J. Qin et al., 2021; Zhan et al., 2021).It was observed that the sources related to vehicular emissions (diesel and petrol driven vehicles), paddy stubble fire, and garbage fire emissions were most contributing factors for ozone formation potential in Mohali (Kumar et al., 2020).*

*Line 652-679*

*In reality, OFP and SOA do not provide complete information about how VOCs influence O3 and organic aerosol chemistry zone formation in Lucknow is more sensitive to NMVOCs concentrations than NOx, similar to other Asian cities. So, Decreasing the VOCs/NOx ratio would also help reduce the secondary pollutants (O3 and SOA). It is observed that vehicular emissions were the main source of aromatics (benzene, toluene, xylene). Therefore, vehicular emission control strategies should be implemented to reduce aromatic (BTEX). Stringent implementation of policies and fuel-efficient standards related to vehicular emissions in Japan and South Korea have primarily improved the air quality (13-17% reduction in NMVOCs) (Wang et al., 2014). In the present study, one of the key observations was that toluene is the main contributor to SOA and ozone production potential. This illustrates that targeting other sources of some NMVOCs (toluene and xylene) will also enhance its control. For example, paint solvents (source of ethylbenzene and xylene) and printing products (source of toluene) were targeted in a city, Hong Kong, where the VOC content of 172 types of consumer products was prescribed by the respective government (Lyu et al., 2017). In the present study, other potential contributor species are methyl cyclohexene (for ozone) and naphthalene (for SOA). These compounds are related to volatile chemical products, as found in the PMF analysis in Lucknow. This infers stringent policies related to solvent-based industries such as textile, automobile, paints, and disinfectants are needed. Regulation and control of NMVOCs content in manufacturing and use of solvent-based products such as pants, disinfectants, fungicides, and insecticides should also be implemented. In China, various industries implemented end-of-pipe measures to control NMVOCs, such as refineries, plant oil extraction, gasoline storage and service stations, pharmacies, and crude oil storage and distribution (Wang et al., 2014) It is also estimated that China's end-of-pipe technologies and new energy-saving policies would help decrease about one-third of NMVOC emissions (Zhang et al., 2020). Phenols and Furans were observed as one of the highest contributors to SOA formation potential related to solid fuel combustion. This suggests controlling solid fuel usage for residential energy and crop-residue burning in the fields within and around the city Lucknow. Firewood burning during the heating period and domestic in-fields straw burning have substantially reduced emissions from biomass burning in China (Wu et al., 2020) . (Derwent et al., 2007) reported that reactivity-based VOC control measures might be more effective than mass-based regulations in controlling ozone and secondary organic aerosol*

*formation. The present study also suggests that the reduction of VOC, especially from vehicular emission is needed for the abatement of ozone and SOA formation in urban areas.*

3) The limitations of these approaches should be noted – these are estimates of potentialfor ozone and SOA formation not actual yields of ozone and SOA.

Reply: we have added the stated limitations in the section.

*Line 643-652*
*These values estimate the potential for ozone and SOA formation and do not indicate the actual yields of ozone and SOA. This estimation method represents the complex behavior of NMVOCs, NOx and solar radiation for producing tropospheric ozone and SOA. There are many NMVOCs species with unknown ozone and SOA yield values. One needs to understand the chemical fates and pathways of many NMVOCs by mimicking real-time atmosphere in smog-chamber studies or through computational modelling studies. More research on this section is needed. Nonetheless, other parameters, including solar radiation and concentration of oxides of nitrogen, also play a key role in the formation of ozone in the troposphere.*

**Conclusion**

1) This section should be used to synthesise what has been learnt from this and the previous studies – what factors are common to Indian/Asian cities and which are different-consider a mapped pie chart type plot for NMVOCs and NR-Pm2.5 composition like that shown in Zhang 2007.

Reply: We thank the reviewer for the comment. In respect to increasing the significance of the manuscript, we added a new section (Section 4) for comparing the study with previous studies in India and Asian cities. We discussed the relevance of the control strategies and common factors in the studies.

*4.     Comparison with other Indian and Asian cities*
*Figure 11 represents mapped pie-chart to compare overall NMVOCs concentrations, and relative source contributions in different Asian and Indian cities. The earlier studies reported the total NMVOCs concentrations between 15-35 ppbv in different cities of China during winters(Wang et al., 2016, 2021a; Hui et al., 2018; Yang et al., 2018). The highest concentration of NMVOCs found in Wuhan city (~34.6 ppbv) with maximum contributions from alkanes and oxygenated VOCs (Hui et al., 2018). The relative composition of sources of NMVOCs found in Wuhan was Industrial/Solvent usage (29.9%), followed by traffic (24.4%), fuel evaporation (23.87%), biomass burning (19.3%) and biogenic (2.5%). The urban site in Beijing reported maximum contribution from secondary VOCs (54.6%), followed by biomass burning (24.4%) and traffic (21%) (Wang et al., 2021a) while the rural site in Beijing had significant contributions from biomass burning (37%) (Yang et al., 2018). Industrial and Traffic contributed similarly at the rural site in Beijing (~31.5%). The difference of source profiles and contributions in urban and rural areas inferred the need of different control strategies and policies in the country (Zhang et al., 2020). It is found that vehicular emissions and biomass burning sources contribute to NMVOCs concentrations (average ~21.5 ppbv) overall 50% and 41% during summers in a land locked urban city, Lhasa, Tibet (Guo et al., 2022) while Industrial/Solvent usage contributed 68% to NMVOCs (average ~33.7 ppbv) in Tokyo, Japan(Fukusaki et al., 2021). It is interesting to note that near the coastal region in Hong Kong, 63.7% and 13.5 % NMVOCs contributions (average ~9.8 ppbv) are related to*

*biomass burning and ship emissions* (Tan *et al.*, 2021). *Despite various air pollution control strategies implemented for more than a decade, NMVOCs and O3 concentrations did not decrease at significant levels in Hong Kong* (Lyu *et al.*, 2017). *Previous study in Kathmandu*(Sarkar *et al.*, 2017), *Nepal demonstrated that biomass co-fired brick kilns (29%) and traffic (28%) contributes to SOA production significantly. Other sources, such as Industrial/ Solvent-usage, biomass burning, and biogenic related emissions also dominated in the city.*

*Earlier source apportionment studies over the NMVOCs mass spectra conducted in Indian cities are limited to two cities in upper IGB region, Delhi (full year) and Mohali (summer). Comparing the urban and sub-urban site of Delhi found that vehicular emissions are dominant at both sites, relatively less contributions to NMVOCs at sub-urban region (36%) as compared to urban region (57%). Throughout the year, traffic emissions dominated the NMVOCs concentration (31%), with comparable contributions from biomass burning (28%), and secondary formations (31%) overall in Delhi. Mohali is located upwind of Delhi city, with maximum contributions from biomass burning (47%), followed by traffic (25%), and secondary formations (16%). The industrial source contributed about 5%, and 12% to NMVOCs concentrations in Delhi and Mohali, respectively. While in the present study, it is found that the solid fuel combustion related emissions majorly (41.3%) contributed to NMVOCs concentrations in Lucknow, located in the central IGB region. The traffic-related emissions (23.5%) and secondary formations (18.6%) are relatively less contributing to NMVOCs as compared to upper IGB region cities (Delhi and Mohali). Moreover, the volatile chemical products emitted more during summer period in Lucknow than compared to Delhi and Mohali. Solid fuel combustion sources aided concentrations of NMVOCs in both Mohali and Lucknow significantly. This may be due to both cities are located downwind of widespread area of agricultural fields. Both of these cities observed relatively less formations of secondary volatile organic compounds, suggested the dominance of fresh emissions than aged compounds in the air mass. Overall, the ambient concentrations of NMVOCs in Indian cities majorly influenced by emissions from solid fuel combustion, vehicular related emissions, secondary formations, and industrial sources. This suggests the need of control measures, and policies implemented for specific sources country-wide and specific to city.*

[Figure]

Figure 11: Mapped- Pie charts showing various sources of NMVOCs in different Asian and Indian cities. The bottom values (in brackets) represent the averaged mixing ratios of total NMVOCs in respective study.

2) Line 546 "The NMVOCs and NOx derive from the formation of ozone and SOA, but there is limited knowledge of their complex relationship" – sentence needs revising. Reverse relationship is true – The formation of ozone and SOA is driven by the oxidation of NMVOCs. The purpose of this statement is not clear.
Reply: We have removed the sentence and improvised the conclusions as given below in the next comment.

3) The overall significance of this work in understanding and better managing air quality inLucknow and other Indian cities needs to be stated.

Reply: Thank you for your suggestion. We have improved the conclusion section as given below. We have included a few studies for different sites in the Delhi region, Ahmedabad, Mohali and Mumbai. However, the measurement of all VOCs measured in this study were not reported for most of the mentioned city. We have included the sentences as per your suggestions.

*Line 739-756*

*The PTR-ToF-MS resolved source factors of NMVOCs were correlated with HR-ToF-AMS resolved factors, Nr-PM2.5 (organics and inorganics), and supporting measurements (BC, NOx, SO2, O3) to analyze their common sources and diurnal patterns. The Ozone and SOA formation potential from individual NMVOCs species and sources were also estimated using MIR and SOA yield values-based methods, respectively. There is a scope for improving these estimates as these values represented the potential for the formation of SOA and O3, not the actual yields. It is found that a few of the NMVOCs species are significantly responsible for secondary pollutant formations. Stringent policies and control actions regarding aromatics (benzene, toluene, xylene, and naphthalene) and oxygenates (phenol and furans) could reduce the NMVOCs emissions drastically. The sources potentially contributing to SOA and ozone formations are traffic, SFC and VCPs. Further control measures and end-to-pipe technologies to reduce emissions from solvent-based industries, consumer products, residential and domestic biomass burning, and vehicular fleets are required to mitigate the health and environmental impacts of NMVOCs and secondary pollutants. The results of this study suggest that to refine the strategies to improve air quality in urban regions of India, particularly the Indo-Gangetic Plain, comprehensive measurements of VOCs are necessary to characterize their emission sources and understand their photochemical processes. This work highlights those local emissions, meteorology, city planning and implementation of the policies in the IGB region highly influence the NMVOCs sources. Further studies focusing on VOCs-secondary organic aerosol interactions would help identify the gas-particle partitioning, ageing and transport of pollution in the region.*

Anonymous Referee #2

The manuscript entitled: "Real-time measurements of NMVOCs in the central IGB, Lucknow, India: Source characterization and their role in O3 and SOA formation" by Jain et al. investigates air quality in Lucknow, India to understand how local and regional emissions contribute to ozone and SOA formation. They employed PMF to apportion NMVOCs to sources related to traffic, solid fuel combustion, volatile chemical products, and secondary formation. These factors were further investigated and found to be consistent with other key measurements such as organic PM PMF factors. Traffic and solid fuel combustion contributed most to SOA formation and OFP. Overall, this article fits the scope of the journal and addresses important questions regarding the sources of pollution which drive air quality. I would recommend publication after major revisions with regards to the comments below.

**General Comments**

Hydrocarbons were measured via PTR-TOF-MS in this study, but alkanes and small alkenes cannot be detected. It is important to be transparent regarding this fact as such species may represent significant fractions of the true total NMVOCs as well as potentially your PMF factors' relative abundance, OFP, and SOA yield. Acknowledgement and discussion of this limitation is necessary.

Reply: We agree with the reviewer that some small compounds, (C1-C4) alkanes and (C1-C4) alkenes, cannot be detected by PTR-ToF-MS. These compounds are substantially present in nature and contribute significant fractions to the true total NMVOCs concentrations. We added some discussion in section 3.2.

*Line 466-472:*
*For the first time, we have included mass spectra of 170 NMVOCs from m/z 42.034 to m/z 197.216 in the PMF analysis. The three abundant NMVOCs (compounds below m/z 42), detected by PTR-ToF-MS, acetaldehyde, acetone, and acetic acid are not included from PMF*

*analysis. Including these NMVOCs in the PMF analysis, resulted in biased solutions where only these ions are well-explained. Additionally, few small alkanes and alkenes (C1-C4) compounds, which are not detected by PTR-ToF-MS are excluded from PMF analysis. However, previous studies have found that these ions are minor contributors to SOA formation. Included compounds (above m/z 42) are major contributors to SOA formation and dominant markers of various sources.*

Key findings regarding factors' relative importance of OFP and SOA yield (and thus, the dominant source(s) of pollution) rely on some significant assumptions (unknown MIRs in Table S2, assumed SOA yields in Table S3). This limitation is briefly noted as "There are many NMVOCs species with unknown ozone and SOA yield values. More research on the section is needed." (lines 529 – 530). It is understandable that the analysis presented here uses the available information, but these limitations must be discussed in greater detail in relation to the findings themselves. Additionally, a sensitivity analysis regarding the unknown values is necessary to substantiate these findings. Do these assumptions make a significant impact or, if not, why?

Reply: We agree with the reviewer that it will bias the result, but the values of MIR of predominant VOCs, which are used as a tracer for different sources in PMF, were taken into account. Therefore, it will not affect the interpretation significantly.

Comparisons of the most abundant NMVOCs, PMF factors, and dominant contributors to OFR and SOA production with other studies of nearby major cities would help put these measurements and conclusions into a greater context.

Reply: We added the comparisons with the previous studies for each aspect as mentioned in different sections. We added one specific section "Comparison with other Indian and Asian cities" to highlight the importance of the study.

*Line 680-720*

*Comparison with other Indian and Asian cities*

*Figure 11 represents mapped pie-chart to compare overall NMVOCs concentrations, and relative source contributions in different Asian and Indian cities. The earlier studies reported the total NMVOCs concentrations between 15-35 ppbv in different cities of China during winters(Wang et al., 2016; Hui et al., 2018; Yang et al., 2018; Wang et al., 2021). The highest concentration of NMVOCs found in Wuhan city (~34.6 ppbv) with maximum contributions from alkanes and oxygenated VOCs (Hui et al., 2018). The relative composition of sources of NMVOCs found in Wuhan was Industrial/Solvent usage (29.9%), followed by traffic (24.4%), fuel evaporation (23.87%), biomass burning (19.3%) and biogenic (2.5%). The urban site in Beijing reported maximum contribution from secondary VOCs (54.6%), followed by biomass burning (24.4%) and traffic (21%) (Wang et al., 2021) while the rural site in Beijing had significant contributions from biomass burning (37%) (Yang et al., 2018). Industrial and Traffic contributed similarly at the rural site in Beijing (~31.5%). The difference of source profiles and contributions in urban and rural areas inferred the need of different control strategies and policies in the country(Zhang et al., 2020). It is found that vehicular emissions and biomass burning sources contributes to NMVOCs concentrations (average ~21.5 ppbv) overall 50%, and 41%, respectively during summers, in a land locked urban city, Lhasa, Tibet(Guo et al., 2022) while Industrial/Solvent usage contributed 68% to NMVOCs (average ~33.7 ppbv) in Tokyo, Japan (Fukusaki et al., 2021a). It is interesting to note that near the coastal region in Hong Kong, 63.7% and 13.5 % NMVOCs contributions (average ~9.8 ppbv) are related to biomass burning and ship emissions (Tan et al., 2021). Despite various air pollution control strategies implemented for more than a decade, NMVOCs and O3 concentrations did not decrease at significant levels in Hong Kong (Lyu et al., 2017). Previous study in Kathmandu(Sarkar et al., 2017), Nepal demonstrated that biomass co-fired brick kilns (29%) and traffic (28%) contributes to SOA production significantly. Other sources, such as Industrial/ Solvent-usage, biomass burning, and biogenic related emissions also dominated in the city.*

*Earlier source apportionment studies over the NMVOCs mass spectra conducted in Indian cities are limited to two cities in upper IGB region, Delhi (full year) and Mohali (summer). Comparing the urban and sub-urban site of Delhi found that vehicular emissions are dominant at both sites, relatively less contributions to NMVOCs at sub-urban region (36%) as compared to urban region (57%). Throughout the year, traffic emissions dominated the NMVOCs concentration (31%), with comparable contributions from biomass burning (28%), and secondary formations (31%) overall in Delhi. Mohali is located upwind of Delhi city, with maximum contributions from biomass burning (47%), followed by traffic (25%), and secondary formations (16%). The industrial source contributed about 5%, and 12% to NMVOCs concentrations in Delhi and Mohali, respectively. While in the present study, it is found that the solid fuel combustion related emissions majorly (41.3%) contributed to NMVOCs concentrations in Lucknow, located in the central IGB region. The traffic-related emissions (23.5%) and secondary formations (18.6%) are relatively less contributing to NMVOCs as*

*compared to upper IGB region cities (Delhi and Mohali). Moreover, the volatile chemical products emitted more during summer period in Lucknow than compared to Delhi and Mohali. Solid fuel combustion sources aided concentrations of NMVOCs in both Mohali and Lucknow significantly. This may be due to both cities are located downwind of widespread area of agricultural fields. Both of these cities observed relatively less formations of secondary volatile organic compounds, suggested the dominance of fresh emissions than aged compounds in the air mass. Overall, the ambient concentrations of NMVOCs in Indian cities majorly influenced by emissions from solid fuel combustion, vehicular related emissions, secondary formations, and industrial sources. This suggested the need of control measures, and policies implemented for specific sources country-wide and specific to city.*

*Line 77-102*
*Introduction*
*Only a few studies have observed and reported the ambient NMVOCs levels in Indian cities. These studies are mainly conducted in large Indian cities such as Delhi (Srivastava, Sengupta and Dutta, 2005; Hoque et al., 2008; Garg, Gupta and Tyagi, 2019; Tripathi et al., 2022), Mumbai (Srivastava, Joseph and Devotta, 2006), Kolkata (Chattopadhyay et al., 1997; Majumdar, Mukherjeea and Sen, 2011), Ahmedabad (Sahu, Yadav and Pal, 2016; Sahu, Tripathi and Yadav, 2017; Tripathi and Sahu, 2020), Udaipur (Yadav et al., 2019; Tripathi et al., 2021), and Mohali (Sinha, Kumar and Sarkar, 2014) .A previous study have presented the health risk assessments for ambient VOCs levels in Kolkata (Majumdar (neé Som) et al., 2008; Chauhan, Saini and Yadav, 2014). Most of these studies have examined only a few NMVOCs, mainly (BTEX), with less or no information related to their sources. Real-time characterization and source apportionment studies for NMVOCs in India are limited to national capital city of Delhi (Wang et al., 2020; Stewart et al., 2021; Jain et al., 2022), and Mohali (Pallavi, Sinha and Sinha, 2019) across different seasons and sites. Traffic emissions and solid fuel combustion are observed to be major contributors in both cities. Significant contributions from secondary VOCs are found in Delhi while solvent based industries contributed to NMVOCs in Mohali. It is necessary to understand the different source profiles and source contributions to ambient NMVOCs in different cities. The atmospheric interactions with radicals, and meteorology highly influence the concentrations of NMVOCs in the region. Recent source apportionment studies based on real-time measurements of non-refractory fine particulate matter using HR-ToF-AMS identified various sources present at different sites in Delhi(Tobler et al., 2020; Lalchandani et al., 2021; Shukla et al., 2021). These studies emphasized that it is essential to understand the variance of sources between day-to-night and different seasons. The major contributors to fine suspended particulate matters in the National Capital Region are the burning of crop residues in neighboring states and open burning of waste, as well as the increased construction activities, industrial expansion, thermal power plants, number of vehicles (two-wheelers and cars), and residential fuel use that result from an ever-increasing population. In addition, recent studies based on real-time measurements of NMVOCs using PTR-ToF-MS in Delhi (Wang et al., 2020; Jain et al., 2022) and Mohali (Pallavi, Sinha and Sinha, 2019) emphasized the importance of source characterization of NMVOCs simultaneously. Very few source apportionment studies highlighted the sources of NMVOCs present in other Asian cities (Sarkar et al., 2017; Hui et al., 2018; Fukusaki et al., 2021b; Tan et al., 2021; Wang et al., 2021). These studies highlighted that NMVOCs sources have substantial value in checking the secondary aerosols formation, and air quality.*

The goal of this work seems to be focused on the main contributors to air pollution, but there is relatively little discussion. Do any individual NMVOCs present a health risk based on your measurements and air quality standards? How frequently do ozone and PM concentrations exceed standards, and which NMVOC and AMS factors seem to drive these events?

Reply: Yes, some of the NMVOCs based on previous epidemiological studies possess great health-risk, which is discussed as given below. We also added the NAAQS standard lines into the timeseries and also added the VOCs species and their source information to the timeseries. It is observed that SFC1 and SFC2 are dominant factors during Jan-Feb. During this period, the PM2.5 exceeds most of the days than the NAAQS standard. PM2.5 exceeds standards more frequently than ozone. Approximately, 80% of the days during the whole study period, PM exceeds the NAAQS standards in the city. We added the discussion as follows in the respective sections.

*Line 400-402*
*During the winter period, the PM2.5 exceeds most of the days than the NAAQS standard. PM2.5 exceed standards more frequently than ozone. Approximately, 80% of the days during the whole study period, PM2.5 exceeds the NAAQS standards in the city, as shown in Figure 2.*

*Line 417-422*
*The CPCB notified the annual National Ambient Air Quality Standards (NAAQS) Only for benzene as 5 (~1.6 ppbv). While WHO recommended no safe level of exposure of benzene. The mean mixing ratio of benzene during the study period found to be 2.9 ±1.9 ppbv which is 2 times higher than the standard guidelines. Prolonged exposure or high short-term exposure to benzene adversely affect the health of citizens of the city due to its haematotoxic, genotoxic and carcinogenic properties.*

[Figure]

**Figure 3: Daily averaged time series of acetaldehyde, acetone, and acetic acid, other NMVOCs, PM₂.₅ and its organic fraction, NO₂, SO₂, O₃, temperature, relative humidity, and wind speed and direction**

**Specific Comments**

Line 18: Notably PTR does not detect alkanes and small alkenes. I suggest specifying "measured" or "quantified" NMVOCs.

Reply: We agree that PTR does not detect alkanes and small alkenes. Therefore, we modified the sentences in the revised manuscript.

*Line 18-20*
*About ~173 NMVOCs from m/z 31.018 to 197.216 were measured during the study period,*

*including aromatics, non-aromatics, oxygenates, and nitrogen-containing compounds.*

Lines 142 – 143: To calibrate the PTR-MS signals, "…a typical value of 2 x 10-9 cm3 s-1 ofthe proton transfer reaction rate coefficient…" was used. Was this for all NMVOCs you did not directly calibrate, or for all NMVOCs you did not have a literature value for? Clarification would be helpful.

Reply. Thanks for your suggestion. We have added a sentence in the revised manuscript.

*Line 191-193*
*The reaction rates (k) of the ions were applied from the literature* (Cappellin *et al.*, 2012)*. "A rate constant of 2 × 10−9 cm3 s−1 was assumed for all ions which reaction rates (k) were not available in the literature (Smith and Spanel, 2005).*

Given that these rate constants vary by a factor of ~2, how does the assumption of an average rate constant affect your calibrations and the rest of your analysis? A discussion on the resulting uncertainties is necessary.

Reply: For the VOCs calibrated using the standard mixture, the overall uncertainties were in the range of 8%–13% in the calculations of the mixing ratios of VOCs including the uncertainties in the mass flow controllers (MFCs) of GCU and standard mixture (±5%–6%). It is also reported in our previous study Sahu & Saxena, 2015; Tripathi & Sahu, 2020. Several studies used k rate 2 × 10−9 cm3 s−1 for species whose reaction rates (k) are not available in the literature. Hansel et al. (1999) estimated the accuracy of the VOC mixing ratio measurements to 30%, mainly caused by the uncertainties of the reaction rate constants, which are up to ±20%. We have discussed these points in the revised version of the manuscript.

Line 193- 199

*The overall uncertainties were in the range of 8%–13% in the calculations of the mixing ratios of VOCs which were present in the standard mixture. The cause of uncertainties in the calculation VOC mixing ratios includes the uncertainties in the mass flow controllers (MFCs) of GCU and standard mixture (±5%–6%). The reaction rates (k) of the ion were applied from the literature (Cappellin et al., 2012). A rate constant of 2 × 10−9 cm3 s−1 was assumed for all ions which reaction rates (k) were not available in the literature.(Hansel et al., 1999; Steinbacher et al., 2004) have reported up to 30% of uncertainty in the calculations of the mixing ratios of VOCs due to k reaction rate.*

What were the sensitivities and limits of detection for your standards? Do estimated LODs have implications for your other measurements (e.g., are your measurements of tetradecane and others above the LOD)?

Reply. The sensitivities of the standards were presented in following Figure. The LOD of the standards in the range of 1 ppt to 45 ppt. Yes, the values below detection limit were removed from the data.

[Figure]

Was the instrument's transmission function investigated / were transmission correction applied?

Reply. Yes, transmission correction was applied.

Since calibrations were done in the beginning, middle, and end of the campaign, were the signals normalized to the reagent ion (as is typical for PTR-MS measurements) to account for relative humidity contributions to the water content in the reactor and general instrument variability?

Reply. Yes, we have already checked the variation of standard (Mentioned in the sensitivity figure) with humidity except few VOCs (Acetone, methanol) most of the VOCs do not show any significant variability with RH.

Were instrument background signals measured and applied? If so, how were they measured?

Reply. Yes, background signals were measured and applied. The detailed information was discussed in our previous studies. As per reviewer 1 suggestion we have included in the SI.

Line 194: There is some inconsistent use of "ions" and "m/z" alongside "mixing ratios." If calibrated NMVOCs were used in your PMF analysis, they are no longer ions. This also applies elsewhere in the manuscript.
Reply: We thank the reviewer for the comment. We checked thoroughly the manuscript and changed the "ions" to the "NMVOCs" or "NMVOC species".

Lines 270 – 271: "…more partitioning of the gas phase during summers relatively to winters." is somewhat ambiguous regarding the direction of partitioning during colder vs warmer months.
Reply: We thank the reviewer for the comment and we have removed the sentence. We also added more information in the section regarding the NMVOCs stats and their relative

contributions in the section.

Line 335: It is unclear to me what "±3" refers to in the context of the scaled residuals.
Reply: When considering scaled residuals as a function of m/z (mass spectra profile) and time (timeseries), the data scatter around zero with the interquartile range almost always between "±3" throughout the entire year, is considered as a good quality of PMF solution. This means that the scaled residuals reasonable range for any m/z or for any particular time period is "±3" (Paatero and Hopke, 2003; Canonaco et al., 2021). Therefore, based on previous literature, we considered the ideal value of "±3" of scaled residuals for each data point when examining the scaled residuals as a function of timeseries for each PMF solution. After that, we finalized the optimum PMF solution. The timeseries graph of scaled residuals (supplementary Figure S4) suggests that the PMF solution with 5-factors is having less than "3" scaled residual values for any given data point. While solutions (2-4 factors) have a lot of data points with more than "3" value of scaled residual.

*Line*
*Another important metric is the evaluation of scaled residuals as a function of timeseries and mass spectra. The scaled residuals ±3 for each data point in the time series are* considered, which is *a evidence of a good quality PMF solution* (Paatero and Hopke, 2003; Canonaco *et al.*, 2021).

Lines 335 – 336: The text would suggest 3 – 7-factor solutions, but Figure S4 shows timeseries for 2 – 10-factor solutions. The diurnals in Figure S4 also include an 11-factor solution whereas the 11-factor timeseries is absent. Figure 4 includes 3 – 11-factor solutions. Please address these inconsistencies.
Reply: We addressed these inconsistencies. We examined the PMF solutions from 3-10 factors solutions. We changed the supplementary Figure and corrected the text also in the main section.

[Figure]

Figure S4: Timeseries and diurnal variation of the scaled residuals from 2-11 factors solution.

Line 339: Estimating from Figure 4, the percent change in Q/Qexp went from ~12% in the 4-factor solution to ~10% in the 5-factor solution to ~8% in the 6-factor solution which does not seem significantly different. Additionally, the total scaled residuals dropped significantly between the 5 and 6 factor solutions. From these parameters alone, one could argue that the 6-factor solution would be better. However, there are other metrics as described previously in the same paragraph that could be used to rule out the 6-factor solution as nonsensical. Please discuss further why the 5-factor solution was chosen as opposed to a 6-factor solution in relation to these other metrics.

Reply: The red line in the Figure 4 represents the % change in Q/Qexp while the black color line and grey shaded area represents the total residuals. The exact values for both parameters related to the respective factor is given below in a table. We have also added this in the supplementary Table. As the 3 to 5 factor solution have maximum % change in Q/Qexp, which is an important parameter. Previous studies have also reported that the solution having more % change in Q/Qexp values will be statistically relevant.

Another parameter, as pointed out, that the total scaled residuals for any given point should be less than 3*standard deviation. So, out of 3, 4 and 5 factor solution, 5-factor solution is having least total scaled residuals. We agree that 6 factor solution is having less total scaled residuals, but it also has less (11% only) d(Q/Qexp) value. After examining the profile, timeseries and durnal of 6 factor solution, one of the factors was split into two factors with

similar profiles.

As pointed out in Figure S4: the timeseries of scaled residuals from 2-10 factor solution., the distinct structure of scaled residual in 4 factor solution timeseries during December while it is not present in the 5-factor solution. This implies that 5 factor solution is better. Moreover, there is no significant change in timeseries when going from 5 factor to 6 factor and so on.

We have also refined the results by constraining with SVOC factor profile. The bootstrap analysis is used to examine the uncertainty in the solution. It is observed that 5 factor solution is having uncertainty of 1% or less for individual each factor as described in the section 2.5. We have also added and restructured the section 2.5 and supplementary text ST1.

| Factor | % Change in Q/Qexp | Total scaled residuals |
|--------|--------------------|------------------------|
| 3 | 20.22 | 142.33 |
| 4 | 19.37 | 104.94 |
| 5 | 18.58 | 81.81 |
| 6 | 11.29 | 68.91 |
| 7 | 9.89 | 62.71 |
| 8 | 12.09 | 55.39 |
| 9 | 8.17 | 52.78 |

Line 363: A short discussion of Figure S5 (as a whole and for each factor) would be helpful for readers such as myself with limited knowledge of this type of analysis. The slopes suggest an error of 1% or less, but the figure seems to tell us more than that. SFC2 seems to have 2 distinct lobes, one with higher concentrations and lower uncertainties (such that PMF latched onto these measurements to determine this factor) and a second lobe with lower concentrations and a higher slope/uncertainty. This effect is stronger for VCPs, and the SVOC factor has essentially no correlation between spread and concentration. Are these observations meaningful and, if so, what do they mean for the quality of the factors?

*Reply: We added some discussion in the supplementary text ST2.*

*Supplementary Text (ST2):*
*The figure represents the analysis of mass error estimation. It is an additional module in advanced ME-2 engine based PMF. As mentioned in supplementary text ST1, we have performed bootstrapping analysis over the mass spectra of our input. Bootstrapping is a technique where; several replicates were generated using resampling strategies. This applied systematic technique aims to explore rotational ambiguity over a defined setting. This analysis gives confidence in extracting the environmentally reasonable PMF runs. The mass error estimation bears information on these PMF runs. For example, in the study, we did bootstrapping for 500 runs, which means 500 times input (mass spectra and time series) were resampled again to check if the same solution with same factors came or not. These factors are defined by same profile and diurnal/ normal cycle. The mass error estimation here suggests the quantifiable error over these 500 PMF runs, when comparing the spread (standard deviation) to its contribution (mean/median). This graph actually suggests the error distribution for each factor for the selected user-criteria based PMF runs. This distribution represented by color schemes could be according to the date (timeseries) or variables (averaged value in the factor). In the graphs, the spread is standard deviation over the mean value for each factor. Given that there are no time dependencies, the relative error can be expressed as a percentage using the slope of the linear fit. In the study, for each factor mass*

*error estimation shows the cumulative average error of 1% for 500 PMF runs with same input after doing bootstrap analysis. This means that the selected solution is very robust, and environmentally reasonable.*

Section 3.2.2: More discussion of the CWT analysis is necessary, including references to Figure 1. Do these results align with expectations?
Reply: we changed the structure of the section 3.2.2 in continuation to the comments by Referee #1. We added the discussion in each sub-section for each factor.

*Line 500-502*
*The back trajectory analysis of the factor, (CWT graph), given in Supplementary Figure S3, shows the probable sources of traffic present near the sampling site.*

*Line 526-530*
*The city is surrounded by various agricultural fields, which generally involve open biomass burning activities. The back trajectory analysis of the factor also shows the probable sources in nearby areas, mainly coming from the west direction of the sampling site (supplementary Figure S3). This argues that this factor is also influenced by the aged biomass-burning plume, transported from sources located on the outskirts of the city and nearby districts.*

*Line 561-563*
*It may be interpreted that SFC 2 is influenced by fresh oxidation of primary biomass burning emissions. Moreover, the CWT plots as shown in supplementary Figure S3, no evidence of its long-range transport is present.*

*Line 600-611*
*This may be due to the influence of particular activity in near-by industries. A conglomerate of the industries is present in the southwest direction of the sampling site within and outside the city, as shown in Figure 1. The direction of the wind changes to the southwest during summers may have brought the high levels of naphthalene and its derivatives emitted from these industrial areas to the sampling site. The CWT graph also shows the strong influence of the source present in the southwest direction of the sampling site (supplementary Figure S3). A previous study has found that among the emitted OVOCs from sewage sludge, first-order OVOCs constituent ~60%, followed by high-order OVOCs* (Haider *et al.*, 2022).*Interestingly, there are three sewage treatment plants located near the sampling site. They may have also influenced the concentrations of OVOCs at the sampling site. The influence of factor contribution during summertime is probably due to the increased production of naphthalene, formaldehyde, and ethanol from their local industrial sources and secondary formations at higher temperatures, as shown in the time series of the factors (supplementary Figure S8).*

There is abundant discussion of key NMVOCs in each factor and how these factors were identified. How do these factors compare to other studies (distribution of NMVOCs within each factor, most abundant NMVOCs, relative abundance of each factor compared to each other, etc.)

Section 3.4: This section briefly reports the results of the OFP and SOA production results, but requires more discussion to articulate the impact and significance of these findings. They should be compared to other studies for context. Limitations should be restated and interpreted with regards to the results.

**Reply:** We thank the reviewer for the suggestions. We improved the section by comparing the results of our study to the previous studies. We also added the discussion regarding the control strategies of ozone and SOA and compared with previous studies of Asian cities. We improved the section replying to comments #1 by Referee 1 (section 3.4), and our Reply is repeated here.

*Line 618-621*

*Toluene, xylene, and isoprene were found to be the highest contributor in terms of OFP in other Asian cities including Guangzhou, and Beijing* (Duan *et al.*, 2008; Zheng *et al.*, 2009; Zhu *et al.*, 2016; Zhan *et al.*, 2021). *In the previous study in Delhi, it has also been noticed that contributions of aromatics (xylene and toluene) have substantital effect to the ozone formation potential* (Tripathi *et al.*, 2022).

*Line 634- 638*

[revised manuscript text omitted]

Lines 513 – 514: When breaking down the OFP (and later SOA) contributions for each factor, one average value is presented for the full measurement period. Do these distributions (and thus dominant factor) vary significantly between seasons (e.g., winter, late winter, and summer as in Figure S1)?

Reply: We added the supplementary figure S3 representing the diurnals from two different seasons, winter and summer for each factor. Each season is divided according to the defined period by IMD, Indian Meteorological Department as winter covering from December to February and summer covering from March to May. As it is observed from the two diurnals, the pattern of variation of sources from day-to-night does not change much between the seasons. However, their relative contributions change significantly. Therefore, We did not divide the period for measuring OFP and SOA, contributions of factors etc. for different seasons. These distributions did not vary between seasons. Only their absolute values differ but overall, their distribution and relative % remain similar.

Figure 5: What is the difference between the bars and dots in the profiles? For the SVOC profile, what do the grey bars represent?
Also, I would suggest creating a new supplementary figure with the factors' diurnals for each season.

Reply: We added the difference between the bars and dots in the profiles in the figure caption itself to be self-explanatory. Also, We added the diurnal for two seasons with the factors in the supplementary Figure 3. These figures show diurnal pattern remains similar for two seasons. However, only difference in their absolute values of NMVOCs concentrations.

*Line 496-497*
*The diurnal pattern is compared between two seasons, winters and summers, also shows similar pattern in supplementary Figure S7.*

[Figure]

**Figure 5: Profile and diurnal variation of individual factors of selected 5-factor solution after PMF analysis at Lucknow for the whole study period. In (a), the left axis represents the relative composition of each factor, given by the vertical bars. The sum of all the bars at different m/z for each factor is 1, and the right axis represents the relative contribution of each factor to a given m/z, shown as grey dots. The grey bars in the SVOC factor represents the degree of constraint on the known source profile and time series. In (b) the middle dark line represents the median of the diurnal while the shaded region represents the interquartile ranges from 25-75[th] percentiles.**

[Figure]

**Figure S3: Diurnal plots of different factors from selected 5-factors solution for different seasons, Winter (Dec-Feb) and Summer (March-May)**

Figure 7a-b: It may be easier to understand this plot with species names as opposed to formulas (or both names and formulas together). Species are already associated with these formulas in Table S1 and "Ethanol" is already provided as a specific compound in this figure. "Unknown" species should be left as formulas.

Reply: We have modified the Figure and added the names and formula for each species in the figure. There are no unknown species.

[Figure]

Figure 10: I believe the third value in each box of (b) and (c) refers to the SOA yield mass concentration and mixing ratios, respectively. Clarification is necessary in the caption. Also, if I am correct, why is the corresponding OFP value not included?

[Figure]

[Figure]

**Figure 10: Distribution in percentage (%) of individual factors to (a) Ozone formation potential (OFP), (b) SOA formation, (c) Relative contribution. The bottom absolute values (in brackets) for (b) and (c) are the SOA yield mass concentration (μg/m3) and average mixing ratios (ppbv)**

Technical Corrections

Line 101: Please define "MSMEs".

Reply: We have added the full form to the text- micro, small and medium enterprises

Line 236: I believe this should be "Equation 5."

Reply: we have made the required changes.

Line 344: Please define "SVOC" (currently defined later in line 368).

Reply: we have made the required changes.

Line 345: Please define "random values" / "a".

Reply: The "a" value is actually a scalar value which varies from 0 to 1. This constraining method can be applied using ME-2 Engine for PMF analysis. In the study, we have used the setting of "a" values starting from 0 to 1 with difference of 0.1 (delta a= 0.1). This means that "a" value would be 0, 0.1, 0.2, 0.3, 0.4, …. till 1. This a- value approach is used to constrain the elements of F matrix (factor profiles) and/ or g matrix (factor time series) as shown in the equation below:

$$f_{j,\ solution} = f_j \pm a \cdot f_j$$

$$g_{i,\ solution} = g_i \pm a \cdot g_i$$

Figure 4: there is no clear link between the traces and their corresponding axes. Add info to the caption, e.g., "…total summed scaled residuals (red trace)…" or change right axis color to red (similar to Figure 3).

Reply: We have made the required changes. In Figure 4, the left axis represents the % change in Q/Qexp which should be red in color. While the total scaled residuals represent black line, filled with grey color. Please find the changed figure below.

[Figure]

Figure 10: Assuming the bottom values for (b) and (c) are the SOA yield mass concentration and average mixing ratios, respectively, please include units

Reply: We have edited the caption of the Figure.
*Figure 10: Distribution in percentage (%) of individual factors to (a) Ozone formation potential (OFP), (b) SOA formation, (c) Relative contribution. The bottom absolute values (in brackets) for (b) and (c) are the SOA yield mass concentration ($\mu g/m^3$) and average mixing ratios (ppbv)*

Figure S1: Please specify the units of wind speed in the legend.

Reply: We have added the units in the caption of Figure S1.

*Figure S2: (a) Wind rose plots showing wind speed (m/sec) and wind direction in different seasons, winter and summer at the sampling site.*

Figure S5: The color schemes seem off. The SVOC factor is colored according to mean concentration (color gradient from left to right), but the others don't follow this same convention. Renaming the color bar label (currently just "ppbv" where both x- and y-axes are in ppbv) could help with interpretation.

Reply: We added the discussing explain the graph and mass error estimation in supplementary text ST2. This distribution represented by color schemes could be according to the date (timeseries) or variables (averaged value in the factor). In this study, we have chosen to represent the color scheme according to the averaged concentration of each factor.

*Supplementary Text (ST2):*

*Mass error estimation for bootstrap analysis:*

*The figure represents the analysis of mass error estimation. It is an additional module in advanced ME-2 engine based PMF. As mentioned in supplementary text ST1, we have performed bootstrapping analysis over the mass spectra of our input. Bootstrapping is a technique where several replicates were generated using resampling strategies. This applied systematic technique aims to explore rotational ambiguity over a defined setting. This analysis gives confidence in extracting the environmentally reasonable PMF runs. The mass error estimation bears information on these PMF runs. For example, in the study, we did bootstrap for 500 runs, which means 500 times input (mass spectra and time series) were resampled again to check if the same solution with same factors came or not. These factors are defined by same profile and diurnal/ normal cycle. The mass error estimation here suggests the quantifiable error over these 500 PMF runs, when comparing the spread (standard deviation) to its contribution (mean/median). The graph (Supplementary Figure S5) suggests the error distribution for each factor for the selected user-criteria based PMF runs. This distribution represented by color schemes could be according to the date (timeseries) or variables (averaged value in the factor). In the graphs, the spread is standard deviation over the mean value for each factor. Given that there are no time dependencies, the relative error can be expressed as a percentage using the slope of the linear fit. In the study, for each factor mass error estimation shows the cumulative average error of 1% for 500 PMF runs with same input after doing bootstrap analysis. This means that the selected solution is very robust, and environmentally reasonable.*

Figure S6: Please define "T/B ratio." It does not seem to be mentioned in the main text.

Reply: We have added the required information in the caption itself of Figure S6.

*Figure S6: Timeseries of high-resolution data for showing particular peaks of industrial events. The T/B ratio represents the ratio of concentrations of toluene/ benzene for the period.*

**References:**

[revised manuscript text omitted]
 2013', Atmospheric Environment, 124, pp. 156–165. doi: 10.1016/j.atmosenv.2015.08.097.*